# Sequential-Parallel Duality in Prefix-Scannable Models

**Morris Yau**[*]
MIT CSAIL
morrisy@mit.edu

**Sharut Gupta**[*]
MIT CSAIL
sharut@mit.edu

**Valerie Engelmayer**
TU Munich
valerie.engelmayer@tum.de

**Kazuki Irie**
Harvard University
kirie@g.harvard.edu

**Jacob Andreas**
MIT CSAIL
jda@mit.edu

**Stefanie Jegelka**
TU Munich, MIT CSAIL
stefje@mit.edu

## Abstract

Modern neural sequence models are designed to meet the dual mandate of parallelizable training and fast sequential inference. Recent developments have focused on various models, such as Gated Linear Attention (GLA) and Mamba, that achieve such "sequential-parallel duality." This raises a natural question: can we characterize the full class of neural sequence models that support near-constant-time parallel evaluation and linear-time, constant-space sequential inference? We begin by describing a broad class of such models, state space models, as those whose state updates can be computed using the classic parallel prefix scan algorithm with a custom associative aggregation operator. We then define a more general class, Prefix-Scannable Models (PSMs), by relaxing the state aggregation operator to allow arbitrary (potentially non-associative) functions such as softmax attention. This generalization unifies many existing architectures, including element-wise RNNs (e.g., Mamba) and linear transformers (e.g., GLA, Mamba2, mLSTM), while also introducing new models with softmax-like operators that achieve $O(1)$ amortized compute per token and $\log(N)$ memory for sequence length $N$. We empirically evaluate such models on illustrative language modeling and canonical synthetic tasks, including state tracking and associative recall. Empirically, we find that PSMs retain the functional effectiveness of transformer-based architectures while matching the inference efficiency of state space models and in some cases exhibiting better length generalization than either.

## 1 Introduction

Transformers have revolutionized sequence processing by enabling parallelizable training over the sequence dimension (Vaswani et al., 2017)—unlike classic recurrent neural networks (RNNs) (Elman, 1990; Jordan, 1986; Hochreiter & Schmidhuber, 1997), which require sequential training; and by handling arbitrary-length sequential dependencies with a constant parameter count—unlike convolutional neural networks, which, while parallelizable over sequence elements, require more parameters to capture longer-range dependencies (Gehring et al., 2017; Oord et al., 2016; Kalchbrenner et al., 2016; Dauphin et al., 2017). However, transformers suffer from two fundamental limitations: first, their computational and memory complexities scale quadratically with sequence length (Vaswani et al., 2017; Katharopoulos et al., 2020), which is particularly problematic during inference; second, they have limited expressivity, i.e., there are computations they struggle to learn to perform (Hahn, 2020; Bhattamishra et al., 2020a;b; Merrill & Sabharwal, 2023; Irie et al., 2023; Merrill et al., 2024; Grazzi et al., 2025; Strobl et al., 2024; Siems et al., 2025; Movahedi et al., 2025).

A body of research in neural sequence modeling has focused on developing architectures that address the primary shortcomings of transformers. In particular, recent years have seen the introduction of diverse models that target the inference time complexity problem. In these models, the inference

---

[*]Equal contribution.

compute requirement is linear in time and constant in memory, just like in classic RNNs, while retaining transformer-like parallelizability during training. Such models include *element-wise recurrent models*, which are derived by simplifying either fully recurrent neural networks (Hochreiter & Schmidhuber, 1997) (e.g. Quasi RNNs (Bradbury et al., 2017) or SRU (Lei et al., 2018); see also (Qin et al., 2023; Li et al., 2018; Balduzzi & Ghifary, 2016; Mozer, 1989) or linear time-invariant dynamical systems (e.g. Mamba (Gu & Dao, 2024))—at the cost of sacrificing expressivity (Merrill et al., 2020; Grazzi et al., 2025). Another model family has been derived from *linear transformers* (Katharopoulos et al., 2020) and *fast weight programmers* (Schmidhuber, 1992; Irie et al., 2021), including Gated Random Feature Attention (Peng et al., 2021), DeltaNet (Schlag et al., 2021; Yang et al., 2024b), RetNet (Sun et al., 2023), GLA (Yang et al., 2024a), mLSTM in xLSTM (Beck et al., 2024), Mamba2 (Dao & Gu, 2024), and versions of RWKV (Peng et al., 2025).

These models share the fundamental property of *sequential-parallel duality* (SPD)—training is parallelizable over sequence elements, while inference is sequential and its inference time complexity is linear. This raises a natural question: *What is the class of neural sequence models that can be evaluated in parallel in nearly constant depth, and sequentially in nearly constant space?*

In this work we aim to characterize the family of models exhibiting SPD. In particular, we show that these models are computable using the classic parallel prefix scan algorithm (Blelloch, 1990; Martin & Cundy, 2018) with a choice of associative aggregation operator that is specific to each model. We define a broader model class, which we call Prefix-Scannable Models (PSMs), by generalizing the aggregation operator used in prefix scan computation. By construction, this family subsumes all existing SPD-compatible models with associative state updates. More generally, it enables the design of novel models with non-associative aggregation rules, whose per-token inference cost remains amortized $O(1)$ with memory scaling $O(\log(N))$ in sequence length $N$. An alternate view is that PSMs are a strict generalization of RNNs: they move beyond affine state updates to support general token mixing operations—including Transformer-style self-attention over local chunks—giving rise to a novel model belonging to the PSM family, which we call Transformer-PSM.

We probe Transformer-PSM in our experiments using small but illustrative tasks: next-token prediction on WikiText-103 (Merity et al., 2017) and synthetic algorithmic tasks that test precise state tracking and retrieval (Merrill et al., 2024; Grazzi et al., 2025; Li et al., 2025; Arora et al., 2024). We find that Transformer-PSMs inherit certain advantages of both Transformers and State Space Models. They preserve the associative recall capability of Transformers, whilst exhibiting an impressive ability to track state. Furthermore, by varying the "chunk" size by which we break up a sequence of tokens, we can alter the asymptotics of a PSM from SSM-like to Transformer-like—a notion we make precise in our discussion on Sequential-Parallel Duality, which we empirically demonstrate on WikiText-103. In summary,

1. We define the SPD family of sequence models and unify modern linear RNNs as those with state computable by the prefix scan algorithm with a custom choice of associative aggregator.

2. We derive a strict generalization thereof, the Prefix Scannable Models (PSMs), that admit general state aggregation functions, such as softmax attention, whilst preserving parallel training in $O(N)$ compute and $O(\log N)$ memory bound at inference.

3. We instantiate Transformer-PSM and evaluate its abilities for state tracking, associative recall, and language modeling, using canonical sequence modeling benchmarks.

## 2 SEQUENCE MODELS AND SEQUENTIAL–PARALLEL DUALITY

Here we formally define sequence models and sequential–parallel duality, and provide examples. For more details on conventions, we refer to Appendix A. Throughout, let $\mathcal{A}$ be a finite alphabet of tokens and $\boldsymbol{a}_{0:n-1} \in \mathcal{A}^n$ an input sequence of length $n$. Let $\mathcal{M}$ be a latent space containing the state of a sequence model. For example, for an RNN, this is the space of the hidden state vector. First, for the sequential view, we define causal sequence models by introducing *state dynamics* and *inference*.

**Definition 2.1** (State kernel). A *state kernel* is a map $U \colon \mathcal{M} \times \mathcal{A} \to \mathcal{M}$ with an identity element $\boldsymbol{e} \in \mathcal{M}$. It induces a *state sequence* $\boldsymbol{s}_{-1} = \boldsymbol{e}$, $\boldsymbol{s}_t = U(\boldsymbol{s}_{t-1}, a_t)$ for $t \geq 0$. We denote by $m(n)$ the memory required to store $\boldsymbol{s}_{n-1}$.

**Definition 2.2** (Inference module). An *inference module* is a map $F \colon \mathcal{M} \times \mathcal{A} \to \mathbb{R}^{|\mathcal{A}|}$ producing a distribution $\hat{\boldsymbol{y}}_t = F(\boldsymbol{s}_{t-1}, a_t)$ over the next token.

**Definition 2.3** (Sequence model). A pair $(U, F)$ comprising a state kernel and an inference module is called a *causal sequence model* (or simply, *sequence model*). The model's *memory bound* $m(n)$ is required to evaluate $U$ and $F$ once the state $\boldsymbol{s}_{n-1}$ is available.

Second, to formalize parallel training, we define a *parallel training circuit* for sequence models.

**Definition 2.4** (Parallel circuit family). A *parallel circuit family* for a sequence model $(U, F)$ is a uniform family of circuits $\{C_n\}_{n \geq 1}$ such that, for all $\boldsymbol{a} \in \mathcal{A}^n$ and all $t < n$, $[C_n(\boldsymbol{a})]_t = F(\boldsymbol{s}_{t-1}, a_t)$, where $\boldsymbol{s}_{t-1}$ is the state (Def. 2.1). The model's *compute bound* $T(n)$ is the size of the circuit $C_n$.

The circuit corresponds to the *training graph*: every token can be processed simultaneously provided sufficient parallel hardware. Together, the sequential and parallel views and their tradeoffs will characterize the Sequential–Parallel Duality (Def. 2.5).

**Definition 2.5** (Sequential–Parallel Duality $\mathsf{SPD}\big(T(n),\, m(n)\big)$). A sequence model $(U, F)$ is said to satisfy $\mathsf{SPD}\big(T(n),\, m(n)\big)$ if the following two conditions hold:

1. **Parallel training.** There exists a uniform circuit family $\{C_n\}_{n \geq 1}$ of *depth* $\tilde{O}(1)$ and *size* $T(n)$ that realises all token-wise predictions (Def. 2.4).

2. **Sequential inference.** Given $\boldsymbol{s}_{t-1}$, the pair $(\boldsymbol{s}_t, \hat{\boldsymbol{y}}_t) = \big(U(\boldsymbol{s}_{t-1}, \boldsymbol{a}_t), F(\boldsymbol{s}_{t-1}, \boldsymbol{a}_t)\big)$ is computable by a depth-$\tilde{O}(1)$ circuit using at most $m(n)$ working memory.

As illustrative examples, we discuss the following sequence models in light of SPD.

**Vanilla Transformer**: $\mathsf{SPD}$-$(n^2, n)$. Training computes all $n^2$ attention scores in parallel with circuit depth $O(1)$ and work $T(n) = \Theta(n^2)$. At inference, each new token requires attending to and storing all $n$ past keys/values, yielding $m(n) = \Theta(n)$ memory.

**Fully recurrent RNN**: *no SPD*. A strict RNN (e.g. LSTM, GRU) updates its hidden state through a chain of length $n$. Because each step depends on the previous one, there is *no* sub-linear-depth circuit that simultaneously computes every output. Such networks therefore fall outside the SPD framework.

As a preview of our results: we will additionally derive the following characterization.

**Prefix–Scannable and Related Models**: $\mathsf{SPD}$-$(n, 1)$ and $\mathsf{SPD}$-$(n, \log(n))$. Modern RNN architectures that admit a Blelloch-style scan (discussed in Sec. 3) for their state update have *compute bound* $T(n) = \Theta(n)$, parallel depth $\Theta(\log n)$, and *memory bound* $m(n) = \Theta(\log n)$ or $m(n) = \Theta(1)$, depending on whether the state size grows logarithmically or remains constant. We therefore write $\mathsf{SPD}$-$(n, \log n)$ or $\mathsf{SPD}$-$(n, 1)$, both of which strictly improve on the Transformer's linear memory latency while retaining fully parallelisable training.

## 3 PREFIX–SCANNABLE MODELS

Next, we define a broad family of models that obtain a sequential-parallel duality of SPD-$(n, \log(n))$. This family consists of sequence models whose *training graph* can be expressed by a *Blelloch prefix scan* (see the caption in Fig. 1) over chunk representations, followed by an independent chunk-local prediction head. The Blelloch scan takes a sequence of tokens or chunks and an aggregation operator, and computes prefixes where the aggregator is applied over the first $n$ tokens; it computes all prefixes in $\Theta(n)$ work and $\Theta(\log n)$ parallel depth. We refer to Alg. 1 in Sec. 3.3 for the full upsweep/downsweep algorithms. We call these *Prefix–Scannable Models* (PSMs). To understand the topic further we first give a brief overview of the classic parallel prefix scan.

**Blelloch Scan.** Let $\mathcal{M}$ be a set with a binary operator $\mathsf{Agg} : \mathcal{M} \times \mathcal{M} \to \mathcal{M}$ and identity $\boldsymbol{e} \in \mathcal{M}$. Given $a_0, \ldots, a_{n-1} \in \mathcal{M}$, the *exclusive prefix* at index $t$ is $P_t := a_0 \, \mathsf{Agg} \, a_1 \, \mathsf{Agg} \, \cdots \, \mathsf{Agg} \, a_{t-1}$ (with $P_0 = \boldsymbol{e}$). The Blelloch prefix–scan computes all $P_t$ in $O(\log n)$ parallel steps via a perfect binary tree: (i) an *upsweep* reduces adjacent pairs bottom-up until the root aggregates the whole sequence; (ii) a *downsweep* propagates prefixes top-down, using stored intermediate values so that every leaf

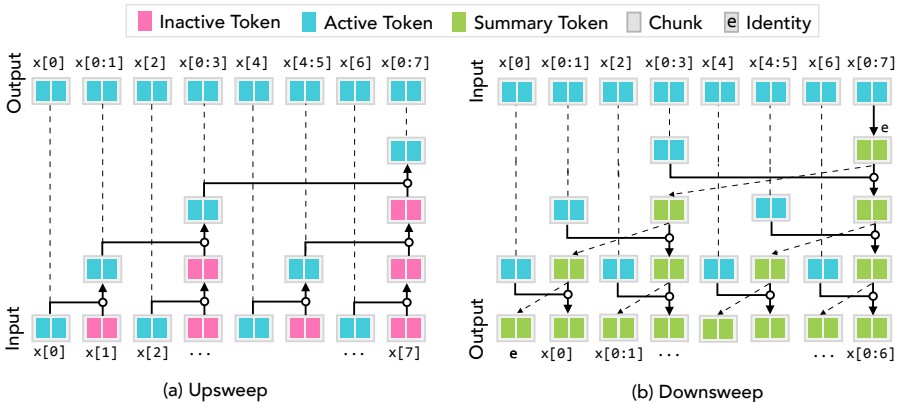

Figure 1: An illustration of the Blelloch parallel scan used to compute prefix states in Prefix-Scannable Models (PSMs). Here the input has 16 tokens grouped into 8 chunks $\{\boldsymbol{x}[0], \ldots, \boldsymbol{x}[7]\}$ (see **(a)** bottom), and the goal is to produce prefix states $\{\boldsymbol{e}, \boldsymbol{x}[0], \boldsymbol{x}[0{:}1], \ldots, \boldsymbol{x}[0{:}6]\}$, where $\boldsymbol{x}[i{:}j]$ aggregates all tokens from chunks $i$ to $j$, and $\boldsymbol{e}$ is the identity. **(a)** In the *upsweep*, chunks are aggregated along a binary tree through a series of chunk aggregation operations (solid arrows), producing intermediate values and some of the final prefix states (e.g., $\boldsymbol{x}[0{:}1], \boldsymbol{x}[0{:}3]$). **(b)** In the *downsweep*, the missing prefix states are filled in by propagating values backward: $\boldsymbol{x}[0{:}7]$ is reset to $\boldsymbol{e}$, and copy (dotted arrows) and aggregation (solid arrows) operations complete the sequence. When each chunk is treated as an atomic element, this recovers the classic Blelloch scan.

receives its $P_t$. When the binary operator Agg is associative, the final prefix array is identical to what a left-to-right sequential loop would compute. If Agg is *not* associative, the result is still well defined—the tree fixes a unique *parenthesisation* (see discussion in Appendix E)—but it may differ from the purely left-nested order used by sequential recurrence. We refer to this upsweep–downsweep as the *static* (training) algorithm (Alg. 1), and to the left-to-right procedure as the *online* (inference) algorithm (Alg. 2), which reproduces the same tree parenthesisation. In the next section we define Prefix Scannable Models (PSMs) by instantiating the static scan with a general choice of Agg.

## 3.1 MODEL DESCRIPTION

**Definition 3.1** (Prefix–Scannable Model). Fix a chunk length $c \leq n$ and partition a sequence $\boldsymbol{a}_{0:n-1}$ into $r = n/c$ disjoint chunks $\boldsymbol{C}_i = (\boldsymbol{a}_{ic}, \ldots, \boldsymbol{a}_{(i+1)c-1})$. A **Prefix–Scannable Model (PSM)** is specified by three learnable modules with depth $O(1)$:

$$\mathsf{Enc} : \mathcal{A}^c \to \mathcal{M}, \quad \mathsf{Agg}_\theta : \mathcal{M} \times \mathcal{M} \to \mathcal{M}, \quad \mathsf{Inf}_\phi : \mathcal{M} \times \mathcal{A}^c \to \mathcal{A}^c,$$

and an identity element $\boldsymbol{e} \in \mathcal{M}$.

1. **Chunk encoding** $\boldsymbol{x}_i = \mathsf{Enc}(\boldsymbol{C}_i)$ for $i = 0, \ldots, r-1$.

2. **Prefix state** $\{\boldsymbol{s}_i\}_{i \in [r]} = \mathrm{BlellochScan}\big(\{\boldsymbol{x}_i\}_{i \in [r]}, \mathsf{Agg}_\theta, \boldsymbol{e}\big)$.

3. **Chunk prediction** $\hat{\boldsymbol{y}}_{ic:(i+1)c-1} = \mathsf{Inf}_\phi(\boldsymbol{s}_{i-1}, \boldsymbol{C}_i)$.

Note that, in terms of notation, we have $\boldsymbol{s}_i = \boldsymbol{x}[0{:}i]$ defined in Fig. 1. We discuss the asymptotics of the PSM model depending on both $n$ and $c$ in Appendix C. For now, we derive the following *immediate complexity corollary* with asymptotics depending on the leading order term $n$, and focus on discussing its connections to recently proposed efficient sequence models. Proposition 3.2 follows from properties of the parallel and streaming versions of the Blelloch scan.

**Proposition 3.2.** *Every Prefix–Scannable Model is in the class* $\mathsf{SPD}\text{-}(n, \log n)$. *That is, its training work is $\Theta(n)$ with parallel depth $\tilde{O}(1)$, while online inference runs in $O(1)$ amortised time and $O(\log n)$ memory per token.*

Table 1: Representative examples of recently proposed layer types that cast into the affine state-update template (Eq. (B.1)). The same associative aggregator $(\boldsymbol{E}, \boldsymbol{f}) \oplus (\boldsymbol{E}', \boldsymbol{f}') \mapsto (\boldsymbol{E} \circ \boldsymbol{E}', \boldsymbol{f} + \boldsymbol{E} \blacktriangleright \boldsymbol{f}')$ is shared by all, and therefore, they are all in $\mathsf{SPD}\text{-}(n, 1)$ by Theorem B.3.

| Model family | $\boldsymbol{E}_t \blacktriangleright \boldsymbol{s}_{t-1}$ | $\boldsymbol{f}_t$ | Gate / operator |
|---|---|---|---|
| Linear Attention (Katharopoulos et al., 2020) | $\boldsymbol{s}_{t-1}$ | $\boldsymbol{v}_t \boldsymbol{k}_t^\top$ | identity $I$ |
| DeltaNet (Schlag et al., 2021) | $\boldsymbol{s}_{t-1}(\boldsymbol{I} - \beta_t \boldsymbol{k}_t \boldsymbol{k}_t^\top)$ | $\beta_t \boldsymbol{v}_t \boldsymbol{k}_t^\top$ | projector |
| Gated DeltaNet (Yang et al., 2025) | $\alpha_t \boldsymbol{s}_{t-1}(\boldsymbol{I} - \beta_t \boldsymbol{k}_t \boldsymbol{k}_t^\top)$ | $\beta_t \boldsymbol{v}_t \boldsymbol{k}_t^\top$ | projector |
| RetNet (Sun et al., 2023) | $\gamma \boldsymbol{s}_{t-1}$ | $\boldsymbol{v}_t \boldsymbol{k}_t^\top$ | scalar gate $\gamma$ |
| mLSTM (Beck et al., 2024) | $f_t \boldsymbol{s}_{t-1}$ | $i_t \boldsymbol{v}_t \boldsymbol{k}_t^\top$ | scalar gate $f_t$ |
| Gated RFA (Peng et al., 2021) | $g_t \boldsymbol{s}_{t-1}$ | $(1 - g_t) \boldsymbol{v}_t \boldsymbol{k}_t^\top$ | scalar gate $g_t$ |
| S4 / S6 (Gu et al., 2022) | $e^{-\boldsymbol{\alpha}_t} \odot \boldsymbol{s}_{t-1}$ | $\boldsymbol{B} \odot (\boldsymbol{v}_t \mathbf{1}^\top)$ | diagonal gate |
| Mamba (Gu & Dao, 2024) | $\bar{A}(\boldsymbol{x}_t) \boldsymbol{s}_{t-1}$ | $\bar{B}(\boldsymbol{x}_t) \boldsymbol{x}_t$ | diagonal gate |
| GLA (Yang et al., 2024a) | $\mathbf{1} \boldsymbol{\alpha}_t^\top \odot \boldsymbol{s}_{t-1}$ | $\boldsymbol{v}_t \boldsymbol{k}_t^\top$ | diagonal gate |

*Proof Sketch.* The static Blelloch scan over $n$ chunk encodings costs linear work and $\Theta(\log n)$ depth (Alg. 1). The streaming evaluation replaces that scan by the online algorithm of Alg. 2, whose Theorem 3.5 and Corollary 3.6 show $O(1)$ amortised work and $O(\log n)$ state. The chunk-local $\mathsf{Inf}_\phi$ adds constant overhead. $\square$

## 3.2 MODERN RNN LAYERS FIT ONE AFFINE SCAN

To relate PSMs to recent models, this section shows that a broad family of recent fast-inference layers (Table 1) are *all* PSM's. Their state kernel can be expressed as specializations of a single associative affine state-update template. This enables $\mathsf{SPD}\text{-}(n, 1)$ complexity.

**Definition 3.3** (Affine recurrence). Let $(\mathcal{M}, +, 0)$ be an additive group and $\blacktriangleright \colon R \times \mathcal{M} \to \mathcal{M}$ a fixed bilinear action of a monoid $(R, \circ, I)$ on $\mathcal{M}$. A layer is said to have an *affine state update* if its hidden state obeys

$$\boldsymbol{s}_t = \boldsymbol{E}_t \blacktriangleright \boldsymbol{s}_{t-1} + \boldsymbol{f}_t, \quad \boldsymbol{s}_{-1} = 0, \tag{3.1}$$

where $(\boldsymbol{E}_t, \boldsymbol{f}_t) \in R \times \mathcal{M}$ are (learnable) functions of the current chunk $x_t$. That is $\boldsymbol{E}_t := \boldsymbol{E}_\theta(\boldsymbol{x}_t)$ and $\boldsymbol{f}_t := \boldsymbol{f}_{\theta'}(\boldsymbol{x}_t)$ for learnable functions $\boldsymbol{E}_\theta$ and $\boldsymbol{f}_{\theta'}$.

The models in Table 1 all satisfy this affine state update template and all share the following associative aggregator. For proof see Appendix B.

**Lemma 3.4.** *(Associative Affine Aggregator) Define for* $(\boldsymbol{E}_i, \boldsymbol{f}_i) \in R \times \mathcal{M}$

$$(\boldsymbol{E}_2, \boldsymbol{f}_2) \oplus (\boldsymbol{E}_1, \boldsymbol{f}_1) = (\boldsymbol{E}_2 \circ \boldsymbol{E}_1, \boldsymbol{f}_2 + \boldsymbol{E}_2 \blacktriangleright \boldsymbol{f}_1), \quad \boldsymbol{e} = (I, 0).$$

*Then* $(R \times \mathcal{M}, \oplus, \boldsymbol{e})$ *is a monoid—$\oplus$ is associative with identity $\boldsymbol{e}$—and*

$$(\boldsymbol{E}_t, \boldsymbol{f}_t) \oplus \cdots \oplus (\boldsymbol{E}_0, \boldsymbol{f}_0) = (\bar{\boldsymbol{E}}_t, \boldsymbol{s}_t),$$

*where* $\boldsymbol{s}_t$ *is the state given by Eq. (B.1) and* $\bar{\boldsymbol{E}}_t$ *is an auxiliary variable.*

Once written in the affine update form, their binary operator is associative, hence each layer is a Prefix–Scannable Model with $\mathsf{SPD}\text{-}(n, 1)$ complexity. For formal theorem and proof see Theorem B.3. Importantly, we can instantiate Def. 3.1 with associative aggregators capturing learnable function families like linear dynamical systems and Gated Linear Attention. Further discussion and the corresponding theorems can be found in Appendix B.1. Next, we turn to general PSM's, which enables new (non-associative) aggregators, most notably softmax attention.

## 3.3 BEYOND AFFINE STATE RECURRENCE: PSMS WITH GENERAL AGGREGATION

The *parallel prefix–scan* computes per-position prefixes with $\mathcal{O}(n)$ work and $\mathcal{O}(\log n)$ depth when the binary operator is associative (Blelloch, 1990). We generalize this view to *non-associative* operators (e.g., softmax attention). For a longer discussion with extensive proofs, see Appendix E.

The key issue is *parenthesisation*: for a non-associative Agg, different groupings of $x_0 \, \text{Agg} \, x_1 \, \text{Agg} \, \cdots \, \text{Agg} \, x_{t-1}$ produce different values. The Blelloch scan resolves this by fixing a single full binary tree (upsweep/downsweep), hence a unique parenthesisation. Let

$$\text{Agg} : \; \mathcal{M} \times \mathcal{M} \to \mathcal{M}, \qquad \text{identity } \boldsymbol{e} \in \mathcal{M}, \tag{3.2}$$

with no associativity assumption unless stated. Define $\pi_{\text{Blelloch}}$ as the binary-tree parenthesisation induced by the static scan. The *static* Blelloch scan (Alg. 1) computes, for every $t$,

$$\boldsymbol{s}_t \;=\; (x_0 \, \text{Agg} \, x_1 \, \text{Agg} \, \cdots \, \text{Agg} \, x_{t-1}) \; \text{ evaluated under } \pi_{\text{Blelloch}}.$$

This matches the sequential left-to-right recurrence when Agg is associative; otherwise it is still well-defined value for the fixed tree. The work is $\mathcal{O}(n)$, and depth is $\mathcal{O}(\log n)$.

**Online binary counter (inference).** The *online* variant (Alg. 2) maintains at most one root per block size $2^k$; inserting $x_t$ performs the usual binary carry with Agg. The current prefix is the most significant bit (MSB) $\to$ least significant bit (LSB) fold of occupied roots. This reproduces *exactly* $\pi_{\text{Blelloch}}$ for each $t$ while using $\mathcal{O}(\log n)$ memory.

---

**Algorithm 1: STATICBLELLOCHSCAN**

**Input:** $\big(\{\boldsymbol{x}_i\}, \text{Agg}_\theta, \boldsymbol{e}\big)$: Array of encoded chunks $\boldsymbol{q}[\boldsymbol{x}_1 ... \boldsymbol{x}_{r-1}]$ with $r = 2^k$ (power of two) chunks; operator Agg with identity $\boldsymbol{e}$

**Output:** Exclusive prefixes written back into $\boldsymbol{q}$

1 **Representation.** Store the complete binary tree in the usual heap layout $T[1 .. 2n-1]$:
    1. leaves $T[n+i] \leftarrow \boldsymbol{q}[i]$ for $i = 0, \ldots, r-1$;
    2. an internal node $v$ has children $2v$ and $2v+1$.

**Upsweep (reduction). for** $v \leftarrow n-1$ **down to** 1 **do in parallel**
    $T[v] \leftarrow \text{Agg}(T[2v], T[2v+1])$

**Downsweep (prefix propagate).** Allocate $P[\,]$; set $P[1] \leftarrow \boldsymbol{e}$ ; // root gets identity
2 **for** $v \leftarrow 1$ **to** $n-1$ **do in parallel**
3     $P[2v] \leftarrow P[v]$;
4     $P[2v+1] \leftarrow \text{Agg}(P[v], T[2v])$

5 **Write back. for** $i \leftarrow 0$ **to** $n-1$ **do in parallel**
6     $\boldsymbol{q}[i] \leftarrow P[n+i]$

---

**Algorithm 2: BINARYCOUNTERUPDATE**

**Input:** $(\texttt{root}, \boldsymbol{x}, \text{Agg}_\theta, \boldsymbol{e})$: Stream of encoded chunks $\boldsymbol{x}_0, \ldots, \boldsymbol{x}_{r-1}$; operator Agg with identity $\boldsymbol{e}$

**Output:** Prefix value $\boldsymbol{p}_t$ for each $t$ (Blelloch parenthesisation)

1 **State:**
2 $\texttt{root}[k]$ stores the root of the current block of size $2^k$ or is $\texttt{empty}$ initialise all to $\texttt{empty}$.

3 **for** $t \leftarrow 0$ **to** $r-1$ **do**
4     $carry \leftarrow \boldsymbol{x}_t$
5     $k \leftarrow 0$
6     **while** $\texttt{root}[k] \neq \texttt{empty}$ **do**
7         $carry \leftarrow \text{Agg}(\texttt{root}[k], carry)$
8         $\texttt{root}[k] \leftarrow \texttt{empty}$
9         $k \leftarrow k+1$
10     $\texttt{root}[k] \leftarrow carry$; // place merged tree
11     $\boldsymbol{p} \leftarrow \boldsymbol{e}$
12     **for** $k \leftarrow \lfloor \log_2(t+1) \rfloor$ **down to** 0 **do**
13         **if** $\texttt{root}[k] \neq \texttt{empty}$ **then**
14             $\boldsymbol{p} \leftarrow \text{Agg}(\boldsymbol{p}, \texttt{root}[k])$
15     **emit** $\boldsymbol{p}$

---

Together, the static and online scans yield PSMs in $\text{SPD}(n, \log n)$: linear work for training and logarithmic memory for streaming inference. (See Fig. 2 for the chunked Transformer-PSM inference.) We obtain the following *correctness and complexity analysis*. We defer proofs to Appendix E

**Theorem 3.5.** *Let $\boldsymbol{p}_t$ be the value emitted at time $t$ by* Alg. 2. *Then $\boldsymbol{p}_t$ equals the exclusive prefix returned by the static Blelloch scan, regardless of whether Agg is associative.*

**Corollary 3.6.** *After $t+1$ chunks* Alg. 2 *stores at most $\lceil \log_2(t+1) \rceil$ root values; hence the worst–case space usage is $\mathcal{O}(\log n)$.*

**Work.** Inserting a new element touches exactly the trailing $1$–bits of $t$; the expected number of such bits is 2, so the amortised number of Agg calls per element is constant.

Together, Theorem 3.5 and Corollary 3.6 show that the online binary–counter scan is an *optimal–space, streamable* realisation of the Blelloch parenthesisation, extending prefix–scan techniques to non–associative operators without increasing asymptotic cost in time. This flexibility enables a larger class of *prefix–scannable models*: sequence models whose per–token state update is any binary operator that admits $\mathcal{O}(\log n)$ space $O(1)$ time online evaluation via the mechanism above. We provide further analytical details of PSMs in Appendix C.

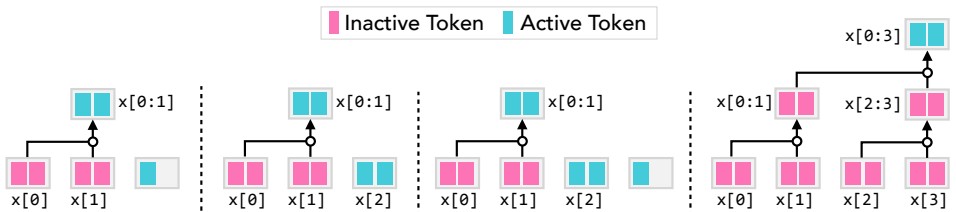

Figure 2: An illustration of the autoregressive state computation of "Transformer-PSM" (Sec. 4) at inference time. Here the model uses a chunk size of 2. From left to right, a single new token is fed to the model at a time. **Two first figures:** when predicting tokens in chunk $x[2]$, the model only requires tokens from the prefix state $x[0{:}1]$ and those within $x[2]$. **Third figure:** predicting tokens in chunk $x[3]$ requires the prefix state $x[0{:}1]$, and chunks $x[2]$ and $x[3]$. **Last figure:** once all tokens in chunk $x[3]$ are processed, a new prefix state $x[0{:}3]$ is computed, which is later used to predict tokens in $x[4]$, and so on. Prefix state $s_i$ corresponds to $s_i = x[0{:}i]$.

## 3.4 TRANSFORMER-PSM

In this section we instantiate Enc, Agg, Inf to concretely define the Transformer-PSM architecture that we use to run our empirics in Sec. 4 to validate our theoretical predictions. The model is uniquely specified by the following three modules.

**Encoder** (Enc): This is a simple embedding layer that transforms discrete vocabulary tokens into continuous vectors, implemented as a standard nn.embedding layer.

**Aggregation** ($\mathsf{Agg}_\theta$). A GPT-2 style Transformer (hidden dim $d$, $H$ heads, $L$ layers) with a *bidirectional* attention mask, $\mathsf{GPT}_\theta^b : \mathbb{R}^{d \times 2c} \to \mathbb{R}^{d \times 2c}$. Given two chunk states $x_i, x_j \in \mathbb{R}^{d \times c}$, define token-concat $[x_i \,|\, x_j] \in \mathbb{R}^{d \times 2c}$ and the *right-half slice* $\mathrm{RH}(Y) \coloneqq Y[:, c:2c] \in \mathbb{R}^{d \times c}$. We write

$$\mathsf{Agg}_\theta(x_i, x_j) \;\coloneqq\; \mathrm{RH}\big(\mathsf{GPT}_\theta^b([x_i \,|\, x_j])\big) \;\in \mathbb{R}^{d \times c}.$$

**Inference** ($\mathsf{Inf}_\phi$). A GPT-2 style Transformer (hidden dim $d$, $H$ heads, $L$ layers) with a *causal* mask, $\mathsf{GPT}_\phi^c : \mathbb{R}^{d \times 2c} \to \mathbb{R}^{d \times 2c}$. Given a prefix state $s_{t-1} \in \mathbb{R}^{d \times c}$ and a token chunk $\mathsf{Enc}(C_t) \in \mathbb{R}^{d \times c}$,

$$\mathsf{Inf}(s_{t-1}, x_t) \;\coloneqq\; \mathrm{RH}\big(\mathsf{GPT}_\phi^c([s_{t-1} \,|\, \mathsf{Enc}(C_t)])\big) \;\in \mathbb{R}^{d \times c},$$

which we interpret as per-token logits for $C_t[1:]$ (next-token prediction within the chunk). Once the three modules are defined, we train Transformer-PSM with Alg. 3 and inference with Alg. 4.

---

**Algorithm 3:** Transformer-PSM Training (static scan over chunks)

**Input:** Sequence of tokens $a_{0:n}$, Enc, $\mathsf{Agg}_\theta$, $\mathsf{Inf}_\phi$, chunk size $c$
**Output:** Predictions $\hat{y}_{0:n}$

1   $r \leftarrow n/c$;    // number of chunks
2   **for** $i \leftarrow 0$ **to** $r$ **do in parallel**
3     $x_i \leftarrow \mathsf{Enc}\big(a_{ic:(i+1)c-1}\big)$
4   $\{s_i\}_{i=0}^r \leftarrow$ STATICBLELLOCHSCAN$\big(\{x_i\}, \mathsf{Agg}_\theta, e\big)$ ;    // Alg. 1
5
6   **for** $i \leftarrow 0$ **to** $r$ **do in parallel**
7     $\hat{y}_{ic:(i+1)c-1} \leftarrow$ $\mathsf{Inf}_\phi\big(s_{i-1}, a_{ic:(i+1)c-1}\big)$

---

**Algorithm 4:** Transformer-PSM Inference (binary-counter scan)

**Input:** Streamed tokens $a_t$, Enc, $\mathsf{Agg}_\theta$, $\mathsf{Inf}_\phi$, chunk size $c$
**Output:** Streaming predictions $\hat{y}_t$

1   **State:**
2    root$[k] \leftarrow$ empty for all $k$ (cf. Alg. 2)
3    buf $\leftarrow [\,]$;    // collects current chunk
4   **for** *each* $a_t$ **do**    // token index $t = 0, 1, \ldots$
5    append $a_t$ to buf;
6    **if** $|buf| = c$ **then**   // completed one chunk
7      $x \leftarrow \mathsf{Enc}(buf)$;
8      $s \leftarrow$ BINARYCOUNTERUPDATE(root, $x$, $\mathsf{Agg}_\theta$, $e$);    // Alg. 2
9      $\hat{y}_{t-c+1:t} \leftarrow \mathsf{Inf}_\phi(s, buf)$;
10      clear buf;

---

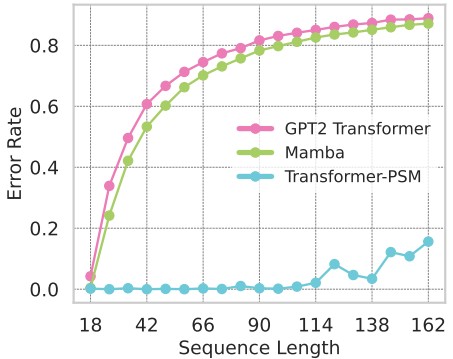 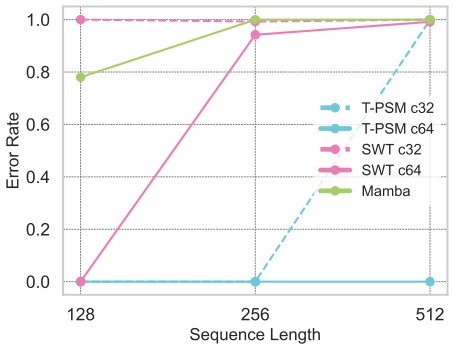

Figure 3: Error rate on the state tracking $S_5$ task. After training on sequences with lengths up to 18, Transformer-PSM generalizes to more than 160 tokens, far beyond Transformer and Mamba.

Figure 4: Error rate on MQAR of Transformer-PSM (T-PSM), Sliding Window Transformer (SWT) and Mamba. Evaluated lengths are in-distribution.

# 4 EXPERIMENTAL RESULTS

The main goal of our experiments is to evaluate and explore the capabilities and properties of Transformer-PSM (Sec. 3.4). For this, we conduct experiments on representative sequence learning tasks: a synthetic algorithmic task requiring state-tracking (Sec. 4.1), a synthetic task for associative recall, and language modeling (Sec. 4.3). Each experiment was conducted on a single NVIDIA V100-32GB GPU. All experiments were implemented using the PyTorch framework (Paszke et al., 2019).

## 4.1 STATE TRACKING $S_5$

The $S_5$ state tracking problem (Merrill et al., 2024; Kim & Schuster, 2023; Li et al., 2025) is the formal version of the "permute cups and balls" challenge, where a sequence of permutations is composed, with the objective of tracking the resulting permutation at each time step. Problems as diverse as tracking finite state automata and evaluation of boolean expressions can be reduced to this task. Naturally, as the sequence of permutations lengthens, this task becomes increasingly difficult for a constant-depth model that has a constant budget for sequential computation. Indeed, the $S_5$ state tracking task is $NC^1$ complete (Barrington, 1986). It is known to be difficult for both standard Transformers and linear RNNs such as Mamba (Merrill et al., 2024; Grazzi et al., 2025).

We train from scratch on sequences of length 4 to 18 in a curriculum and subsequently evaluate on lengths up to 180 to test for length generalization. We generate 100,000 sequences per length and train for 20 epochs for each of three different models: (1) a standard GPT2 model with 12 layers, 12 heads, 768 hidden dimensions; (2) a 370M-parameter Mamba model with 48 layers and a 1024-dimensional hidden state; (3) Transformer-PSM with ($d = 768, H = 1, L = 1$) for Agg, ($d = 768, H = 1, L = 1$) layer for Inf, and chunk size $c = 1$. All models are trained with Adam with dropout 0.1, weight decay 0.01, learning rate $10^{-4}$.

Fig. 3 shows the results. We find that whilst Mamba slightly outperforms GPT2, the new T-PSM has remarkably low error rate even for sequences significantly longer than those observed during training, showing that these models exhibit strong length generalization for state tracking tasks.

## 4.2 MULTI QUERY ASSOCIATIVE RECALL (MQAR)

In Associative Recall, the task is to recall whatever value followed a key earlier in a given sequence. MQAR extends this task to multiple such key-value pairs to increase the memory demand (Arora et al., 2023). While constant state size recurrent models struggle with this task, a 2-layer transformer excels by solving it perfectly. To gauge where on this spectrum our model falls, we train different models on MQAR for 64 epochs with vocabulary size 8192 and 8 key-value pairs. In the typical setting of this task, sequences are constructed in a way that a key is queried shortly after it appears for the first time; here we do not use such a bias and sample queries uniformly, which makes the task harder than the standard setting.

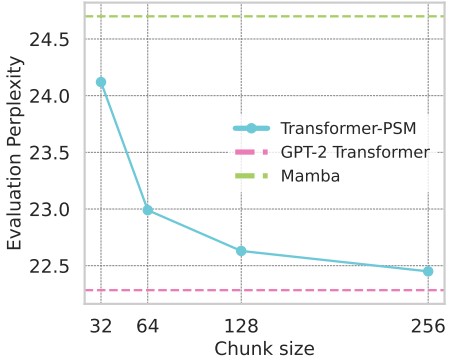
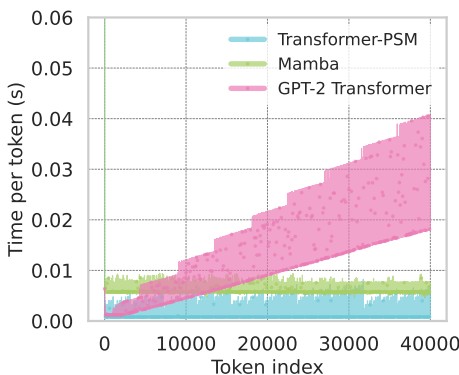

Figure 5: Evaluation perplexity of Transformer-PSM with varying chunk sizes on WikiText-103

Figure 6: Inference time per token for Transformer-PSM and GPT-2 Transformer

Here we instantiate Transformer-PSM with $(d = 256, H = 1, L = 2)$ Agg, $(d = 256, H = 1, L = 2)$ Inf. We also use a learnable linear projection to compress the chunks instead of taking the right half. The chunk size is 32 or 64. For comparison, we include both Mamba and Sliding Window Transformer (SWT) baselines (Beltagy et al., 2020; Zaheer et al., 2020). The SWT is a GPT2 model with $(d = 256, H = 1, L = 4)$, where we use a sliding window size of 32 or 64.

Fig. 4 shows the results. Here all the evaluation lengths are in the training distribution. We find that T-PSM with a chunk size of 64 achieves the perfect accuracy like the full context transformer, while reducing its chunk size to 32 yields performance degradation on a long length (512). Mamba fails in our setting; unlike in prior work (Arora et al., 2024; Okpekpe & Orvieto, 2025), our setting is harder due to our uniform query sampling as discussed above.

### 4.3 Language Modeling on WikiText-103 with Transformer-PSM

Here we evaluate perplexity on the WikiText-103 dataset (Merity et al., 2017). We benchmark Transformer-PSM $(d = 768, H = 12, L = 1)$ Agg, $(d = 768, H = 12, L = 11)$ Inf, by varying the self-attention chunk size from 32 to 256 tokens and measuring test perplexity against the vanilla GPT-2 (base) baseline with a context size of 512 trained from scratch. As shown in Fig. 5, as the chunk size grows, perplexity falls gracefully from 24.12 at 32 tokens to 22.45 at 256 tokens—closely approaching vanilla GPT-2's perplexity of 22.28—demonstrating that larger chunks recover nearly full-context modeling power while preserving our model's linear-time inference. For reference, we also include a baseline for Mamba trained from scratch at $130m$ parameters, 768 hidden dimension, trained for 10 epochs with the same optimizer hyperparameters achieving a ppl of 24.7.

Next, we measure per-token latency over 40,000 WikiText-2 tokens for our model versus a 4-layer, 4-head, 256-dimensional GPT-2 baseline. We train Transformer-PSM $(d = 768, H = 4, L = 2)$ Agg, $(d = 768, H = 4, L = 2)$ Inf, thus keeping the parameter count identical to the baseline. As shown in Fig. 6, GPT-2's inference cost grows linearly with context length ($O(n)$ per token) with KV cache, inflating latency from $\approx 0.002s$ at the start to $\approx 0.04s$ by token 40,000. In contrast, our Transformer-PSM design reuses 64-token chunk summaries, leading to a $O\big(2n + \frac{n}{32} \log(n/64)\big)$ inference cost (as discussed in Eq. (C2) in Appendix D), keeping per-token time below $\approx 0.008s$. For reference, we also include inference time measurement for a Mamba model with 4 layers, 256 hidden dimension, with an average inference time per token of $\approx .006s$.

### 5 Discussion and Conclusion

**Discussion.** We give a concise conceptual view of parallelisable, inference-efficient sequence models via *prefix scannability*, unifying many closely related models developed under different names. Our results deepen the link between prefix-scan algorithms and efficient sequence models, extending the design space beyond prior work (Martin & Cundy, 2018).

This algorithmic lens offers a framework for analysing and designing future models. For example, concurrent work on "log linear attention" (Guo et al., 2025) also fits this view, proposing a

linear-attention mechanism with $\log n$ memory, structured state, and an efficient chunkwise-parallel primitive.

**Conclusion.** We formalise *sequential–parallel duality*: models that train in parallel yet decode sequentially. Recent efficient sequence models exhibit this duality and achieve linear-time inference. We characterise them as instances of the classic parallel prefix-scan with a model-specific operator, motivating and analysing the broader class of *parallel scannable models* (PSMs). In particular, we go beyond existing examples of PSMs by defining and empirically studying a novel sequence model based on *non-associative* aggregators. Our experiments suggest that such model may have benefits in length generalization for some tasks, and opens avenues of exploring this design space in light of specific applications. Overall, this provides an insightful unification of efficient sequence models, that cannot be found in any prior work.

## ACKNOWLEDGEMENTS

This project was partially supported by NSF award CCF-2112665, the MIT Quest for Intelligence, and the Alexander von Humboldt Foundation. S.G. is supported by the MathWorks Engineering Fellowship.

## REPRODUCIBILITY STATEMENT

We provide proofs in Appendix B and relevant background for all theoretical results in Sec. 2 and Appendix A. For experiments, we detail the training protocols in Sec. 4, and algorithm implementations in Sec. 3.4. All datasets are publicly available, and we follow established preprocessing procedures. We will release all code and scripts in a public GitHub repository upon acceptance.

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

APPENDIX

## A    PRELIMINARIES AND DEPTH CONVENTIONS

**Depth conventions.** Throughout this paper we restrict attention to *causal sequence models whose training and inference graphs can be executed, in the Random–Access Machine (RAM) model with unbounded fan-out gates, at depth (measured by the longest path of synchronous operations)*

$$O\big(\mathrm{polylog}\, n\big), \quad \text{abbreviated } \tilde{O}(1).$$

Whether the hidden polylog factor is $\log n$ or $(\log n)^2$ hinges on the chosen primitive set—for instance, treating GEMM and `softmax` as unit-time kernels versus expanding them into arithmetic gates. Classical fully recurrent networks such as LSTMs (Hochreiter & Schmidhuber, 1997) and GRUs (Cho et al., 2014), whose forward pass has depth $\Theta(n)$ and therefore admits *no* sub-linear parallel schedule, fall outside this scope.

Our focus is on *polynomial separations* between the principal model families: standard Transformers, prefix-scannable models (as we will define in Sec. 3), and linear recurrent RNNs. Unless a logarithmic factor is essential to the argument, we suppress it with the tilde notation. To characterize our example models, we need to specify how their depth is counted, i.e., in the model of computation: How tall is their training circuit?

**Transformer.** A single self–attention head executes the composite map $(\boldsymbol{Q}, \boldsymbol{K}, \boldsymbol{V}) \mapsto \mathrm{softmax}\big(\boldsymbol{Q}\boldsymbol{K}^\top/\sqrt{d}\big)\boldsymbol{V}$, where $d$ denotes the key/query head dimension. In a Random–Access Machine with unbounded fan-out gates, the *pointwise* linear projections $\boldsymbol{x}\boldsymbol{W}$ have depth 1, but the $n \times n$ *matrix multiply* $\boldsymbol{Q}\boldsymbol{K}^\top$ and every row-wise *softmax* (vector sum + normalisation) require a parallel reduction of $n$ numbers. Using a binary tree this costs $\Theta(\log n)$ depth.[*] Hence, an $L$–layer Transformer has depth $D(n) = \Theta(L \log n)$. If one *treats the GEMM and softmax kernels as unit-time primitives*, this is often reported as "constant depth," but strictly speaking it is $\tilde{O}(1)$ (polylogarithmic).

**Mamba, Gated Linear Attention, RWKV.** The expensive step is a parallel scan that produces the running state. Its depth is $\Theta(\log n)$, and the pointwise gating around it adds $O(1)$. Stacking $L_{\mathrm{agg}}$ such layers gives $D(n) = \Theta\big(L_{\mathrm{agg}} \log n\big)$.

## B    ADDITIONAL PROOFS

### B.1    MODERN RNN LAYERS FIT ONE AFFINE SCAN

To relate PSMs to recent models, this section shows that a broad family of recent fast-inference layers (Table 1) are *all* specializations of a single affine state-update template, i.e. their state kernel can be expressed as an affine bilinear function. This enables SPD-$(n, 1)$ complexity.

**Definition B.1** (Affine recurrence). Let $(\mathcal{M}, +, 0)$ be an additive group and $\blacktriangleright: R \times \mathcal{M} \to \mathcal{M}$ a fixed bilinear action of a monoid $(R, \circ, I)$ on $\mathcal{M}$. A layer is said to have an *affine state update* if its hidden state obeys

$$\boldsymbol{s}_t = \boldsymbol{E}_t \blacktriangleright \boldsymbol{s}_{t-1} + \boldsymbol{f}_t, \quad \boldsymbol{s}_{-1} = 0, \tag{B.1}$$

where $(\boldsymbol{E}_t, \boldsymbol{f}_t) \in R \times \mathcal{M}$ are (learnable) functions of the current chunk $x_t$. That is $\boldsymbol{E}_t := \boldsymbol{E}_\theta(\boldsymbol{x}_t)$ and $\boldsymbol{f}_t := \boldsymbol{f}_{\theta'}(\boldsymbol{x}_t)$ for learnable functions $\boldsymbol{E}_\theta$ and $\boldsymbol{f}_{\theta'}$.

The models in Table 1 all satisfy this affine state update template and all share the following aggregator, which is associative.

**Lemma B.2.** *(One associative operator, Affine aggregator) Define for* $(\boldsymbol{E}_i, \boldsymbol{f}_i) \in R \times \mathcal{M}$

$$(\boldsymbol{E}_2, \boldsymbol{f}_2) \oplus (\boldsymbol{E}_1, \boldsymbol{f}_1) = \big(\boldsymbol{E}_2 \circ \boldsymbol{E}_1, \ \boldsymbol{f}_2 + \boldsymbol{E}_2 \blacktriangleright \boldsymbol{f}_1\big), \quad e = (I, 0).$$

*Then* $(R \times \mathcal{M}, \oplus, \boldsymbol{e})$ *is a monoid—$\oplus$ is associative with identity $\boldsymbol{e}$—and*

$$(\boldsymbol{E}_t, \boldsymbol{f}_t) \oplus \cdots \oplus (\boldsymbol{E}_0, \boldsymbol{f}_0) = \big(\bar{\boldsymbol{E}}_t, \ \boldsymbol{s}_t\big),$$

*where* $\boldsymbol{s}_t$ *is the state given by Eq. (B.1) and* $\bar{\boldsymbol{E}}_t$ *is an auxiliary variable.*

---

[*] There are sub-logarithmic circuits for exact matrix multiply (e.g. Valiant, 1975), but they are very wide and rarely exploited in ML practice; $\log n$ therefore matches realistic GPU / TPU kernels.

*Proof.* Straightforward verification using the action axioms; full details in Appendix B. □

Once written in that form, their binary operator is associative, hence each layer is a Prefix–Scannable Model with SPD-$(n, 1)$ complexity.

**Theorem B.3.** *Every layer that satisfies Def. B.1 is a Prefix–Scannable Model with chunk size $c = 1$, encoder* Enc, *aggregator* Agg, *and inference module* Inf *defined as*

$$\boldsymbol{x}_i = \mathsf{Enc}(\boldsymbol{C}_i) = (\boldsymbol{E}_i, \boldsymbol{f}_i)$$

*for $i = 0, \ldots, r - 1$ with the aggregator of Lemma B.2.*

$$\mathsf{Agg}(\boldsymbol{x}_2, \boldsymbol{x}_1) := (\boldsymbol{E}_2, \boldsymbol{f}_2) \oplus (\boldsymbol{E}_1, \boldsymbol{f}_1)$$

*We define* Inf$(\boldsymbol{s}_{i-1}, \boldsymbol{C}_i)$ *to be the function that takes states and current token and outputs predictions. Hence these models admits training work $\Theta(n)$, parallel depth $\Theta(\log n)$, and online inference cost $O(1)$ time and $O(1)$ memory per token: the layer is in* SPD-$(n, 1)$.

*Proof.* Apply the static Blelloch scan to the pairs $(\boldsymbol{E}_i, \boldsymbol{f}_i)$ using $\oplus$ to obtain every prefix in $O(n)$ work and $O(\log n)$ depth. Lemma B.2 ensures the scan outputs the correct state $\boldsymbol{s}_t$, which the inference head may consume chunk-wise. During streaming inference, the online left to right scan maintains the same prefixes with constant work and constant additional memory because $\oplus$ is associative. □

Table 1 shows a catalogue of affine layers. Note that the affine form absorbs normalisation variables common in linear Transformers (e.g. running scalars/vectors $z_t$; typically running sum of keys (Katharopoulos et al., 2020) or related variables (Beck et al., 2024)) by enlarging the state vector and treating the auxiliary variable as just another coordinate updated affinely. The proof of Theorem B.3 requires no change.

## B.2 Examples of Prefix-Scannable Sequence Models

In the following, we present two families of models whose parallel circuits can be obtained as the computation of a Blelloch parallel scan. In fact, it suffices to show that for all family of architectures that are affine in their state, there exists an associative operator $\oplus$ that defines a monoid over which the Blelloch parallel scan operates.

One type of prefix-scannable models are **linear time invariant dynamical systems**.

**Definition B.4** (LTI Linear Dynamical System). A linear time invariant system is defined by four matrices $(\boldsymbol{A}, \boldsymbol{B}, \boldsymbol{C}, \boldsymbol{D}) \in \mathbb{R}^{d \times d}$ defining

$$\boldsymbol{s}_{t+1} = \boldsymbol{A}\boldsymbol{s}_t + \boldsymbol{B}\boldsymbol{x}_t \tag{B.2}$$
$$\boldsymbol{y}_t = \boldsymbol{C}\boldsymbol{s}_t + \boldsymbol{D}\boldsymbol{x}_t \tag{B.3}$$

Here $\boldsymbol{s}_0 = 0$ is the initial state, and $\boldsymbol{s}_t$ is the state at time $t \in \mathbb{Z}^+$. $\boldsymbol{x}_t \in \mathbb{R}^d$ is the input at time $t$.

**Definition B.5** (Associative Operator for Affine State Monoid). For each timestep, let $\boldsymbol{g}_t$ be an augmented pair $\boldsymbol{g}_t := (\boldsymbol{E}_t, \boldsymbol{f}_t) := (\boldsymbol{A}, \boldsymbol{B}\boldsymbol{x}_t)$ where $\boldsymbol{E}_t \in \mathbb{R}^{d \times d}$ is a matrix and $\boldsymbol{f}_t \in \mathbb{R}^d$ is a vector. We define an associative operator $\oplus$ as

$$(\boldsymbol{E}_2, \boldsymbol{f}_2) \oplus (\boldsymbol{E}_1, \boldsymbol{f}_1) = (\boldsymbol{E}_2\boldsymbol{E}_1, \boldsymbol{f}_2 + \boldsymbol{E}_2\boldsymbol{f}_1) \tag{B.4}$$

To demonstrate that a sequence model is Prefix Scannable, we must verify two properties. Firstly, that the operator $\oplus$ applied to all the $g_i$ over all timesteps computes the state. Secondly that, $\oplus$ is associative.

**Lemma B.6.** *Let $\boldsymbol{G}_t$ be the augmented pair equal to $\oplus$ applied to the sequence of augmented pairs $\boldsymbol{g}_1, ..., \boldsymbol{g}_T$. Then*

$$\boldsymbol{G}_t = (\boldsymbol{E}_t, \boldsymbol{f}_t) \oplus ... \oplus (\boldsymbol{E}_1, \boldsymbol{f}_1) = (\boldsymbol{A}^t, \sum_{k=0}^{t-1} \boldsymbol{A}^{t-1-k}\boldsymbol{B}\boldsymbol{x}_k) \tag{B.5}$$

*Secondly for any inputs $\boldsymbol{g}_i, \boldsymbol{g}_j, \boldsymbol{g}_k$ we have $(\boldsymbol{g}_i \oplus \boldsymbol{g}_j) \oplus \boldsymbol{g}_k = \boldsymbol{g}_i \oplus (\boldsymbol{g}_j \oplus \boldsymbol{g}_k)$*

*Proof.* Proof by induction for the equality and straightforward computation for associativity.

We have base case.
$$(\boldsymbol{E}_2, \boldsymbol{f}_2) \oplus (\boldsymbol{E}_1, \boldsymbol{f}_1) = (\boldsymbol{A}^2, \boldsymbol{B}\boldsymbol{x}_2 + \boldsymbol{A}\boldsymbol{B}\boldsymbol{x}_1) \tag{B.6}$$
Apply definitions to see this is true for general $t$.

Proof of associativity.

$\boldsymbol{g}_1, \boldsymbol{g}_2, \boldsymbol{g}_3$ we have $(\boldsymbol{g}_3 \oplus \boldsymbol{g}_2) \oplus \boldsymbol{g}_1 = \boldsymbol{g}_3 \oplus (\boldsymbol{g}_2 \oplus \boldsymbol{g}_1)$

$$\boldsymbol{g}_3 \oplus (\boldsymbol{g}_2 \oplus \boldsymbol{g}_1) = (\boldsymbol{A}, \boldsymbol{B}\boldsymbol{x}_3) \oplus (\boldsymbol{A}^2, \boldsymbol{B}\boldsymbol{x}_2 + \boldsymbol{A}\boldsymbol{B}\boldsymbol{x}_1) = (\boldsymbol{A}^3, \boldsymbol{B}\boldsymbol{x}_3 + \boldsymbol{A}\boldsymbol{B}\boldsymbol{x}_2 + \boldsymbol{A}^2\boldsymbol{B}\boldsymbol{x}_1) \tag{B.7}$$
$$= (\boldsymbol{A}^2, \boldsymbol{B}\boldsymbol{x}_3 + \boldsymbol{A}\boldsymbol{B}\boldsymbol{x}_2) \oplus (\boldsymbol{A}, \boldsymbol{B}\boldsymbol{x}_1) = (\boldsymbol{A}, \boldsymbol{B}\boldsymbol{x}_3) \oplus (\boldsymbol{A}, \boldsymbol{B}\boldsymbol{x}_2) \oplus (\boldsymbol{A}, \boldsymbol{B}\boldsymbol{x}_1) \tag{B.8}$$
$$= (\boldsymbol{g}_3 \oplus \boldsymbol{g}_2) \oplus \boldsymbol{g}_1 \tag{B.9}$$
$\square$

Another type of prefix-scannable models are **linear transformers** and their gated variants.

**Definition B.7.** Gated Linear Attention (GLA) is defined with a states $\boldsymbol{s}_1, ..., \boldsymbol{s}_T \in \mathbb{R}^{p \times d}$, inputs $\boldsymbol{x}_1, ..., \boldsymbol{x}_T$, gating function $r : \mathbb{R}^d \to \mathbb{R}$, keys $\boldsymbol{k}_1, ..., \boldsymbol{k}_T$, kernel map $\phi : \mathbb{R}^d \to \mathbb{R}^p$

$$\boldsymbol{s}_t = r(\boldsymbol{x}_t) \odot \boldsymbol{s}_{t-1} + \phi(\boldsymbol{k}_t)\boldsymbol{v}_t^T \tag{B.10}$$
$$\tag{B.11}$$

We observe that GLA has an affine state recurrence.

**Lemma B.8.** *Let $\boldsymbol{E}_t \in \mathbb{R}$ be a scalar that can be computed from $\boldsymbol{x}_t$. Let $\boldsymbol{f}_t \in \mathbb{R}^{p \times d}$ be a matrix that can be computed from $\boldsymbol{x}_t$. Then the GLA state recurrence is an affine function of the form*

$$\boldsymbol{s}_t = \boldsymbol{E}_t \boldsymbol{s}_{t-1} + \boldsymbol{f}_t \tag{B.12}$$
$$\tag{B.13}$$

*In particular, let $\boldsymbol{g}_t = (\boldsymbol{E}_t, \boldsymbol{f}_t)$ be an augmented pair, and let $\oplus$ be an operator defined as*
$$(\boldsymbol{E}_2, \boldsymbol{f}_2) \oplus (\boldsymbol{E}_1, \boldsymbol{f}_1) = (\boldsymbol{E}_2\boldsymbol{E}_1, \boldsymbol{f}_2 + \boldsymbol{E}_2\boldsymbol{f}_1) \tag{B.14}$$
*Then $\oplus$ is associative, and $\boldsymbol{s}_t = \boldsymbol{g}_t \oplus ... \oplus \boldsymbol{g}_1$.*

*Proof.* Proof by induction for the equality and straightforward computation for associativity.

First we prove $\boldsymbol{s}_t = \boldsymbol{g}_t \oplus ... \oplus \boldsymbol{g}_1$ by induction. Consider the base case.
$$\boldsymbol{g}_2 \oplus \boldsymbol{g}_1 = (r(\boldsymbol{x}_2), \phi(\boldsymbol{k}_2)\boldsymbol{v}_2^T) \oplus (r(\boldsymbol{x}_1), \phi(\boldsymbol{k}_1)\boldsymbol{v}_1^T) \tag{B.15}$$
$$= (r(\boldsymbol{x}_2) \odot r(\boldsymbol{x}_1), \phi(\boldsymbol{k}_2)\boldsymbol{v}_2^T + r(\boldsymbol{x}_2) \odot \phi(\boldsymbol{k}_1)\boldsymbol{v}_1^T) \tag{B.16}$$
$$= (r(\boldsymbol{x}_2) \odot r(\boldsymbol{x}_1), \boldsymbol{s}_2) \tag{B.17}$$
Then assuming the identity holds at timestep $t - 1$
$$\boldsymbol{g}_t \oplus (r(\boldsymbol{x}_{t-1}) \odot ... \odot r(\boldsymbol{x}_1), \boldsymbol{s}_{t-1}) = \tag{B.18}$$
$$(r(\boldsymbol{x}_t) \odot ... \odot r(\boldsymbol{x}_1), r(\boldsymbol{x}_t) \odot \boldsymbol{s}_{t-1} + \phi(\boldsymbol{k}_t)\boldsymbol{v}_t^T) \tag{B.19}$$
as desired.

Then we also check associativity.

$$\boldsymbol{g}_3 \oplus (\boldsymbol{g}_2 \oplus \boldsymbol{g}_1) = \boldsymbol{g}_3 \oplus (r(\boldsymbol{x}_2) \odot r(\boldsymbol{x}_1), \phi(\boldsymbol{k}_2)\boldsymbol{v}_2^T + r(\boldsymbol{x}_2) \odot \phi(\boldsymbol{k}_1)\boldsymbol{v}_1^T) \tag{B.20}$$
$$= (r(\boldsymbol{x}_3) \odot r(\boldsymbol{x}_2) \odot r(\boldsymbol{x}_1), \phi(\boldsymbol{k}_3)\boldsymbol{v}_3^T + r(\boldsymbol{x}_3) \odot \phi(\boldsymbol{k}_2)\boldsymbol{v}_2^T + r(\boldsymbol{x}_2) \odot r(\boldsymbol{x}_2) \odot \phi(\boldsymbol{k}_1)\boldsymbol{v}_1^T) \tag{B.21}$$
$$= (r(\boldsymbol{x}_3) \odot r(\boldsymbol{x}_2), \phi(\boldsymbol{k}_3)\boldsymbol{v}_3^T + r(\boldsymbol{x}_3) \odot \phi(\boldsymbol{k}_2)\boldsymbol{v}_2^T) \oplus \boldsymbol{g}_1 \tag{B.22}$$
$$= (\boldsymbol{g}_3 \oplus \boldsymbol{g}_2) \oplus \boldsymbol{g}_1 \tag{B.23}$$
$\square$

## C  ANALYTICAL DETAILS OF CHUNKING IN PREFIX SCANNABLE MODELS

Here we summarize the analytical details of PSMs (Def. 3.1). The model state of a PSM after $t$ chunks is the Blelloch prefix defined to be $s_t = \mathrm{Agg}_\theta^{\mathrm{Blelloch}}(a_0{:}a_t)$, and the model outputs $\hat{y}_t = \mathrm{Inf}_\phi(s_{t-1}, C_t)$. The same state sequence $\{s_t\}$ can be produced *online* with $O(\log t)$ memory by replacing the static scan with the binary-counter scan of Alg. 2. The corresponding $\mathcal{O}(\log t)$-parallel depth training algorithm and $O(\log t)$-memory online decoding algorithm can be found in Alg. 3 and Alg. 4, respectively. Note that both parallel loops and the STATICBLELLOCHSCAN have depth $\mathcal{O}(\log t)$, dominated by the static Blelloch scan, so the whole training pass admits efficient batch execution.

The model has the following properties:

**Sequential–parallel duality.** Alg. 3 and Alg. 4 produce *identical* state sequences $\{s_t\}$ (Theorem 3.5), so a PSM trained with the static scan can be evaluated online with logarithmic memory.

**Model family.** Choosing $\mathrm{Agg}_\theta$ to be associative recovers known scan-friendly models as a strict subset of PSMs; non-associative choices (e.g. a Transformer block) enlarge the design space while retaining online decodability.

**Complexities.** Training: For sequences of length $n$, chunks of size $c$, we have $\mathcal{O}(n)$ work, $\mathcal{O}(\log(n/c))$ depth. Online inference: $\mathcal{O}(c)$ amortised work per token and $\mathcal{O}(c\log(n/c))$ memory after $n/c$ chunks.

Further details about the computational complexity are detailed below in Appendix D.

## D  COMPUTATIONAL COMPLEXITY OF PSMS

Let

- $n$ – sequence length,
- $c$ – chunk size ($n = c \cdot num\_chunks$),
- $L_{\mathrm{agg}}$ – number of Transformer layers in $\mathrm{Agg}_\theta$,
- $L_{\mathrm{inf}}$ – number of Transformer layers in $\mathrm{Inf}_\phi$,
- $d_s$ and $d_x$ – hidden widths of the two modules (held constant).

Throughout we count one forward–backward pass as a single "time unit" and use the usual dense-attention cost $\mathcal{O}(L\,\ell^2 d)$ for a length-$\ell$ Transformer block with $L$ layers. Only the *scaling with* $n, c, L_{\mathrm{agg}}, L_{\mathrm{inf}}$ is retained; constant factors in $d_s, d_x$ are suppressed.

**Training (static Blelloch scan).** The three parallel loops of Alg. 3 give

$$T_{\mathrm{train}} = \mathcal{O}(cnL_{\mathrm{agg}} + cnL_{\mathrm{inf}}), \qquad S_{\mathrm{train}} = \mathcal{O}\Big(cn\,L_{\mathrm{inf}} + cnL_{\mathrm{agg}}\Big). \tag{C1}$$

*Depth* is $\mathcal{O}\big(L_{\mathrm{inf}} + \log(n/c)\,L_{\mathrm{agg}}\big)$. because the static Blelloch scan dominates parallel runtime. Total nonparallel runtime is linear in sequence length $O(cnL_{\mathrm{agg}} + cnL_{\mathrm{inf}})$. Additional factor of $c$ comes from $c^2$ dense attention for $n/c$ chunks.

**Online inference (binary-counter scan).** Each new chunk incurs

1. one $\mathrm{Inf}_\phi$ call $\Rightarrow$ cost $\mathcal{O}(L_{\mathrm{inf}}\,c^2)$, and
2. at most $\log(n/c)$ calls to $\mathrm{Agg}_\theta$ per chunk $\Rightarrow$ amortised cost $\mathcal{O}(L_{\mathrm{agg}})$.

Hence, for the whole length-$n$ stream, we make $n$ calls to $\mathrm{Inf}$ and $\frac{n}{c}$ calls to $\mathrm{Agg}$. The space at inference is to store the kv-cache for the $c$ tokens in $\mathrm{Inf}$ and the $\log(n/c)$ chunks of $c$ tokens in $\mathrm{Agg}$

$$T_{\text{infer}} = \mathcal{O}\Big(nc\,L_{\text{inf}} + nc\,L_{\text{agg}}\Big), \qquad S_{\text{infer}} = \mathcal{O}\Big(c\,L_{\text{inf}} + c\log(n/c)\,L_{\text{agg}}\Big). \tag{C2}$$

**Per-token latency.** Dividing (C2) by $n$ gives

$$\mathcal{O}\Big(c\,L_{\text{inf}} + c\,L_{\text{agg}}\Big)$$

work and $\mathcal{O}(\log n)$ space, confirming constant-time amortised decoding under fixed $c$.

**Remarks.**

- When $c = \Theta(1)$ (token-wise chunks) both training and inference are linear in $n$ with *constant* memory for $\mathsf{Inf}_\phi$ and logarithmic memory for $\mathsf{Agg}_\theta$.
- For larger $c$ the quadratic self-attention of $\mathsf{Inf}_\phi$ over each chunk dominates work.
- If $\mathsf{Agg}_\theta$ is associative, we may swap the static and online scans without affecting costs; thus SSMs and gated linear attention inherit (C1)– (C2) as special cases.

## E  BEYOND AFFINE STATE RECURRENCE, PSM'S WITH GENERAL AGGREGATION: FURTHER DETAILS

The *prefix–scan* (a.k.a. parallel prefix) is fundamental to many parallel algorithms. When the binary operator is *associative*, the classic Blelloch scan (Blelloch, 1990) computes, in $\mathcal{O}(n)$ work and $\mathcal{O}(\log n)$ depth, the same left–to–right prefix values that a sequential loop would produce. This section extends the view to *non–associative* operators such as those expressible by softmax attention.

But, there is a challenge with non-associativity: the numerical results of straightforward parallel and sequential versions would differ since parenthesisation differs, challenging our duality. *Parenthesisation* here means the explicit placement of parentheses that fixes *which two elements are combined first* when evaluating a long chain of binary operations. For instance,

$a\,\mathsf{Agg}\,b\,\mathsf{Agg}\,c\,\mathsf{Agg}\,d$   may be grouped as   $((a\,\mathsf{Agg}\,b)\,\mathsf{Agg}\,c)\,\mathsf{Agg}\,d$ or $a\,\mathsf{Agg}\,(b\,\mathsf{Agg}\,(c\,\mathsf{Agg}\,d))$,

and when Agg is *not* associative the two expressions generally differ. The Blelloch algorithm removes this ambiguity by committing to a single, fixed parenthesisation: the full binary tree generated by its upsweep and downsweep. All variants we describe—static and online—evaluate *exactly that same tree*, guaranteeing identical results even for non-associative operators.

We first review the static tree formulation, then present an online variant that realises *exactly the same parenthesisation* while using only $\mathcal{O}(\log n)$ memory. Throughout, let

$$\mathsf{Agg} \;:\; \mathcal{M} \times \mathcal{M} \;\rightarrow\; \mathcal{M}, \qquad \text{identity element } \boldsymbol{e} \in \mathcal{M}, \tag{A1}$$

be an arbitrary binary operator. No associativity assumption is required unless stated.

First, we introduce the static Blelloch scan which is a "parallel" training over sequence elements. Then we introduce the online binary counter scan which is the "sequential" inference over sequence elements that computes prefixes with the same parenthesisation. The runtime required to run the static Blelloch scan is $T(n) = O(n)$, whereas the amount of space required during the online binary counter scan is $m(n) = O(\log(n))$. Taken together this analysis gives us PSMs in SPD-$(n, \log(n))$ i.e linear compute during training and nearly linear space during inference.

**Static Blelloch Scan (Alg. 1).** Alg. 1 is agnostic to Eq. (A1), that is, it is valid for *any* operator. When Agg is not associative, however, the output for index $t$ no longer equals the sequential recurrence $\boldsymbol{s}_t = \mathsf{Agg}(\boldsymbol{a}_t, \boldsymbol{s}_{t-1})$. Instead, it is the unique value obtained by applying Agg along the fixed binary–tree parenthesisation imposed by the algorithm. The next subsection shows how to realise *the same parenthesisation* online with logarithmic space.

**Online Binary–Counter Scan (Alg. 2).** The online variant processes the stream $a_0, \ldots, a_{n-1}$ left to right while maintaining a *binary counter* of complete mini–trees. At time $t$ (0–indexed) the binary

expansion of $t+1$ reveals which block sizes $2^k$ are present. There is at most one mini–tree (its root value) per block size, hence at most $\lceil\log_2(t+1)\rceil$ roots in memory. Inserting a new element is identical to adding 1 to a binary counter: each trailing `1` bit triggers a merge with Agg and a carry to the next bit. Aggregating the occupied roots from most- to least-significant bit (MSB $\to$ LSB) reproduces the value that the static Blelloch tree would hold after processing the same prefix.

We obtain the following **correctness and complexity analysis**.

**Proposition E.1.** *After processing $t+1$ elements ($t \geq 0$), every non–empty* `root[k]` *equals the aggregate of the* most–recent $2^k$ *tokens* $\boldsymbol{x}_{t-2^k+1}, \ldots, \boldsymbol{x}_t$, *and* $(t+1)$ *is divisible by* $2^k$.

*Proof.* By induction on $t$. The base case $t=0$ is immediate. For the inductive step, the carry chain merges two adjacent blocks of size $2^k$ precisely when bit $k$ flips from `1` to `0` in the binary counter. The merged value therefore covers the $2^{k+1}$ most recent tokens and is stored at position $k+1$, where divisibility holds. Untouched positions keep their invariant. $\square$

**Theorem 3.5.** *Let* $\boldsymbol{p}_t$ *be the value emitted at time $t$ by* Alg. 2. *Then* $\boldsymbol{p}_t$ *equals the exclusive prefix returned by the static Blelloch scan, regardless of whether Agg is associative.*

*Proof.* By Proposition E.1 the occupied roots partition the first $t+1$ tokens into blocks whose sizes are decreasing powers of two when listed MSB $\to$ LSB. This is exactly the leaf order of the perfect binary tree used by the static algorithm. Each block's internal value was constructed by the same sequence of merges as in that tree; aggregating the blocks left–to–right therefore reproduces the tree's evaluation order and thus its numeric result. $\square$

**Corollary 3.6.** *After $t+1$ chunks* Alg. 2 *stores at most $\lceil\log_2(t+1)\rceil$ root values; hence the worst–case space usage is $\mathcal{O}(\log n)$.*

*Proof.* The binary representation of $t+1$ contains at most $\lfloor\log_2(t+1)\rfloor+1$ bits, and there is at most one root per bit. $\square$

**Work.** Inserting a new element touches exactly the trailing `1`–bits of $t$; the expected number of such bits is 2, so the amortised number of Agg calls per element is constant.

Together, Theorem 3.5 and Corollary 3.6 show that the online binary–counter scan is an *optimal–space, streamable* realisation of the Blelloch parenthesisation, **extending prefix–scan techniques to non–associative operators without increasing asymptotic cost in time**. This flexibility enables a larger class of *prefix–scannable models*: sequence models whose per–token state update is any binary operator that admits $\mathcal{O}(\log n)$ space $O(1)$ time online evaluation via the mechanism above. We provide further analytical details of PSMs in Appendix C.

