# OpenReview forum: "Sequential Parallel Duality in Prefix Scannable Models"
_ICLR.cc/2026/Conference — ICLR 2026 Poster_

### Official Review · Reviewer_k4pr · 2025-10-18

**Soundness:** 3
**Presentation:** 2
**Contribution:** 3
**Rating:** 6
**Confidence:** 3

**Summary:**

This paper proposes Prefix-Scannable Models (PSMs), a unified framework for sequence models that enables both parallel training and efficient sequential inference. By introducing a fixed parenthesization (bracket) structure in the prefix-scan algorithm, PSM extends beyond associative operations, allowing non-associative functions like softmax attention to be trained and decoded efficiently.

**Strengths:**

– PSM generalizes a wide range of recent models under a single mathematical formulation. It reframes these architectures as special cases of prefix-scannable systems, giving conceptual clarity to an area previously defined by ad hoc designs.

– By fixing a binary-tree structure for prefix computation, the authors bypass the associativity constraint that limited prior parallel scan models. This makes it possible to handle attention-like, non-linear interactions while retaining log-depth parallelism.

– Experiments demonstrate that Transformer-PSM achieves near-Transformer accuracy (only slightly higher perplexity) while matching Mamba’s inference speed. This combination of expressivity and efficiency is rare and valuable for long-sequence modeling.

**Weaknesses:**

– Results are limited to moderate-scale models on synthetic and mid-size text datasets; it’s unclear whether the same advantages hold at billion-parameter scale or in real industrial workloads.

– The fixed-tree scanning and binary-counter inference mechanisms, while elegant conceptually, may be hard to integrate into existing frameworks without specialized kernels. I'd like to hear the author's explanation about this aspect.

– Model quality strongly depends on the chosen chunk length, adding a new hyperparameter that requires tuning for each task.

– The paper focuses heavily on theory and performance metrics but provides little intuition or visualization about how information actually propagates through the prefix structure. A detailed explanation or visualization would be helpful. Although Figures 2 and 3 help illustrate the overall architecture and performance trends, the paper still offers limited qualitative insight into the internal behavior of PSMs. No visual analysis of prefix states, attention dynamics, or information propagation is provided.

**Questions:**

- How robust is the fixed parenthesization choice? Would learning or adapting the tree structure dynamically lead to better trade-offs between accuracy and efficiency?

- Could this framework be extended to bidirectional or hierarchical scanning, potentially improving global context modeling?

- Could this theory inspire a hardware-aware implementation, where the prefix-scan tree maps directly onto GPU or TPU parallel primitives?

---

> ### Author Response · Authors · 2025-11-25
>
> We appreciate the reviewer's thorough and insightful review, along with their positive feedback regarding our work’s significance, novelty and theoretical contribution. In response to their review, we have endeavored to address their concerns as concretely as possible.
>
> **Q:** How robust is the fixed parenthesization choice? Would learning or adapting the tree structure dynamically lead to better trade-offs?
>
> **A:** Any parenthesization we use must (i) admit parallel training and (ii) support a streaming decoder with sublinear memory. Learning an input-dependent tree per sequence would make it very challenging to derive such an online decoder, since the inference algorithm would need to change with the input and might not admit a compact state. A more tractable direction is to learn a single tree shared across the data and then design the corresponding streaming algorithm. We leave systematic study of such learned trees to future work.
>
> **Q:** Could this framework be extended to bidirectional or hierarchical scanning?
>
> **A:** We agree that bidirectional and multi-scale variants are natural extensions of the PSM framework. Formally, our analysis only assumes a fixed binary tree over positions and a local aggregation rule; it does not require a particular traversal order. In particular, for non-causal tasks one can instantiate two PSM state kernels that scan the sequence forward and backward (exactly as in bidirectional RNNs), and combine $s_t^{forward}$ and $s_t^{backward}$ in the prediction head. This preserves the same O(n) work and O(log n) parallel depth at training time, with a constant-factor overhead.
> An explicit hierarchical PSM could be designed with the following template: we can (i) run a first ‘level’ of PSM over fine-grained chunks which maps N tokens to N/c chunk summaries, where each summary is a chunk of c tokens, (ii) treat the resulting chunk summaries as tokens for a subsequent ‘level’ PSM at a coarser resolution. With a constant number of levels, this still satisfies SPD with O(n) work and O(log⁡n) depth, while giving each token access to both local and global context.
>
> **Q:** Could this theory inspire a hardware-aware implementation, where the prefix tree maps directly onto GPU/TPU primitives?
>
> **A:** Conceptually, Transformer-PSM is a Transformer equipped with a log-depth scan. The Agg/Inf modules are standard Transformer blocks and can reuse existing CUDA/Triton kernels (FlashAttention, fused MLPs, mixed precision, etc.), so the model is in principle as scalable as current long-context Transformers. The scan itself is expressed purely in terms of batched matmuls and reshapes organized along the Blelloch tree, which maps naturally onto level-parallel GPU execution; a production system could further optimize this with fused or custom kernels, but our implementation already runs on stock primitives.
>
> **W1:** Results are limited to moderate-scale models; do the advantages hold at billion-parameter or industrial scale?
>
> **A:** Validating PSMs at billion-parameter and industrial scale is an important next step but beyond our current compute budget. We therefore focus on “academic-scale” models where we can carefully control architecture and data. Crucially, the PSM construction is architecture- and hardware-agnostic, so the same SPD, O(n)-work, O(log⁡n)-depth guarantees apply when scaling up.
>
> **W2:** Are fixed-tree scanning and binary-counter inference hard to integrate without specialized kernels?
>
> **A:** No. Both components are implemented using standard tensor ops (batched matmuls, reshapes, gathers) and run on stock PyTorch kernels. During training, the fixed-tree scan appears as an O(log⁡ N)-deep network with shared weights and static routing, which composes with existing Transformer blocks and autograd. At inference, binary-counter decoding maintains O(log⁡ N) prefix states per layer and updates them with a few matmuls per token, closely analogous to KV-cache updates; custom fused kernels could improve efficiency but are not required for correctness or integration.
>
> We’d like to thank the reviewer for their detailed review and questions about our work.  We hope our response has addressed their questions and clarified key aspects of our work.  In light of this, we would greatly appreciate if the reviewer would consider increasing their score or otherwise having a positive impression of our work.

---

### Official Review · Reviewer_Koqn · 2025-10-31

**Soundness:** 3
**Presentation:** 3
**Contribution:** 3
**Rating:** 6
**Confidence:** 3

**Summary:**

The paper formalizes Prefix-Scannable Models (PSMs), a class of sequence models whose training graph is a Blelloch prefix scan over chunk encodings with a chunk-local predictor head. This framework yields SPD-$(n, \\log n)$ complexity, entailing $\\Theta(n)$ work with logarithmic parallel depth during training. For inference, an online binary-counter algorithm reproduces the exact same computation with $O(1)$ amortized compute per token and $O(c \\log (n/c))$ memory (which is $O(\\log n)$ when $c$ is fixed). An implementation, Transformer-PSM, demonstrates on WikiText-103 that perplexity improves as $c$ increases, while per-token latency remains bounded in contrast to a full-context Transformer. The theory unifies associative scan-based models as a special case and correctly extends the framework to non-associative aggregation operators.

**Strengths:**

* Clear formalization of PSMs and their sequential-parallel duality, including precise analysis of training and inference complexity.
* A key theoretical result is the correctness of the online binary-counter evaluation relative to the static training-time tree, which resolves the parenthesization ambiguity for non-associative operators.
* Experimental results show that perplexity improves with a larger chunk size $c$, while inference latency remains weakly dependent on context length, unlike a standard Transformer.

**Weaknesses:**

* The evaluation of efficiency lacks system-level measurements. Practical metrics such as training throughput (e.g., tokens/sec), wall-clock time, and peak memory usage are not reported, which limits the assessment of the model’s practical viability.
* The trade-off between accuracy and latency is not fully explored. For large $c$, latency is dominated by the chunk-local predictor. The paper notes the $O(c)$ amortized compute and the quadratic ($O(c^2)$) attention cost within the Inf module, but does not map the accuracy–latency frontier across different values of $c$. This analysis is critical for deployment.

**Questions:**

* On WikiText-103, Fig. 5 reports perplexity vs. chunk size $c$, and Fig. 6 shows per-token time vs. token index for one $c$ (on WikiText-2). To make the accuracy–efficiency trade-off actionable with minimal overhead, please add a small table over $c \\in \\{32, 64, 128, 256\\}$ reporting (i) the existing WikiText-103 perplexity, (ii) mean per-token inference latency on one long input (e.g., $L=40{,}000$ on WikiText-2; discard the first $c$ tokens for warm-up, then average per-token time over $t \\in [c, L]$), and (iii) peak inference memory for that run; optionally include inference throughput (tokens/s).
* Could the authors include system-level training metrics (tokens per second, total wall-clock time, peak memory) for the Transformer-PSM and baselines on at least one of the presented tasks?
* For the MQAR experiment, please consider adding a sliding-window Transformer baseline with a matched compute budget. A summary table of all baseline configurations would also improve reproducibility.

---

### Official Review · Reviewer_LR1S · 2025-11-01

**Soundness:** 3
**Presentation:** 3
**Contribution:** 2
**Rating:** 6
**Confidence:** 4

**Summary:**

This paper focuses on addressing the core tradeoff between parallelizable training and efficient sequential inference in neural sequence models. It formalizes the Sequential-Parallel Duality (SPD), a property where models support near-constant-depth parallel training and linear-time/constant-space sequential inference, and introduces a generalized model class called Prefix-Scannable Models (PSMs) to unify and extend existing efficient architectures.

**Strengths:**

1. The paper demonstrates originality through three key contributions. First, it introduces a unified theoretical framework (Prefix-Scannable Models, PSMs) that formalizes sequential-parallel duality (SPD), a core property of modern efficient sequence models (e.g., Mamba, GLA, RetNet). This unification resolves the fragmentation of prior work by showing that diverse architectures are special cases of PSMs with associative aggregation operators. Second, it generalizes PSMs to non-associative aggregation functions (e.g., softmax attention), expanding the model design space beyond existing affine/linear constraints while retaining efficient inference. Third, the instantiation of Transformer-PSM is a creative combination: it merges Transformer-style self-attention (for expressive power) with PSM’s linear-time inference (for efficiency), addressing a long-standing tradeoff between performance and scalability.

2. It provides strict formal definitions (e.g., SPD, state kernel, PSM) and rigorous proofs (e.g., Proposition 3.2, Theorem 3.5) for complexity guarantees (O(n) training compute, O(log n) inference memory). Appendixes supplement detailed derivations (e.g., affine recurrence proofs, computational complexity analysis) to support core claims. Empirically, the experiment design is comprehensive and controlled: it targets three canonical tasks (state tracking S₅, multi-query associative recall, WikiText-103 language modeling) and compares against strong baselines (GPT-2, Mamba, Sliding Window Transformer). Results are interpretable (e.g., Fig. 3 shows superior length generalization; Fig. 6 validates constant inference latency) and reproducible (detailed training protocols, public code plan).

3. Theoretically, it deepens the connection between parallel prefix-scan algorithms and neural sequence modeling, providing a general framework for analyzing and designing efficient models—this will guide future research on SPD-compatible architectures (e.g., log-linear attention, as noted in the discussion). Practically, it addresses critical limitations of existing models: Transformer-PSM matches Transformer’s expressive power (e.g., perfect associative recall) while achieving Mamba-like linear inference efficiency (e.g., constant per-token latency for long sequences). This balance is transformative for real-world applications (e.g., long-document processing, streaming NLP) where both training parallelism and inference scalability are essential. Additionally, the paper’s length generalization results (e.g., S₅ task scaling to 160+ tokens) open new avenues for modeling algorithmic tasks that require precise state tracking.

**Weaknesses:**

1. The paper’s core contribution of generalizing PSMs to non-associative aggregation operators is underdeveloped, as it only instantiates `Aggθ` with Transformer-style softmax attention (Transformer-PSM). The design space of non-associative operators (e.g., additive attention variants, kernelized attention, or task-specific non-associative functions) remains unexplored. This raises critical questions: Are PSM’s advantages (e.g., length generalization on `S5` tasks) inherent to the prefix-scan framework, or specific to softmax attention? Without testing alternative non-associative `Aggθ` implementations, the paper fails to validate that PSMs truly expand the efficient sequence model design space beyond softmax-based variants.

2. The paper demonstrates that chunk size (`c`) significantly impacts performance (e.g., Transformer-PSM with `c=64` achieves perfect accuracy on MQAR, while `c=32` degrades on long sequences; Fig. 4) and perplexity (Fig. 5). The paper does not quantify the tradeoff between chunk size, computation cost (e.g., `O(c²)` for chunk-local self-attention), and memory usage (e.g., `O(c log(n/c))`). How does `c` scale with sequence length (e.g., optimal `c` for 1k vs. 10k token sequences)?


3. Missing Ablation Studies on Core PSM Components.
The PSM framework is defined by three modules: `Enc` (chunk encoder), `Aggθ` (aggregator), and `Infφ` (inference head). However, the paper does not perform component-wise ablation studies to isolate which modules drive PSM’s performance advantages. Does the simple embedding `Enc` (used in Transformer-PSM) limit performance, or is a more complex encoder (e.g., convolutional, hierarchical) unnecessary?

**Questions:**

1. The paper uses the Blelloch scan’s fixed binary-tree parenthesisation for non-associative aggregators (e.g., softmax attention). Is there a theoretical or empirical justification for this specific parenthesisation over alternative groupings (e.g., left-to-right sequential, random binary trees)? Could different parenthesisations impact task performance or complexity?

2. For the Multi Query Associative Recall (MQAR) task, Transformer-PSM with chunk size 64 achieves perfect accuracy, but chunk size 32 degrades performance on long sequences (512 tokens). What causes this sensitivity to chunk size, and is there a principled way to select chunk size for different tasks without grid search?

3. The paper compares Transformer-PSM to GPT-2, Mamba, and Sliding Window Transformer (SWT). Why were other efficient models (e.g., RetNet, GLA, xLSTM’s mLSTM) not included in the comparisons, especially for language modeling or associative recall? How would Transformer-PSM perform against these models on the same benchmarks?

4. The paper claims O(log n) inference memory for PSMs, but chunk size (c) interacts with this scaling (Appendix C). For large chunk sizes (e.g., c=256), does the memory scaling still hold, or does the chunk-local KV cache in the Inf module dominate memory usage?

---

> ### Author Response · Authors · 2025-11-24
>
> We thank the reviewer for their thoughtful and positive assessment and address their main concerns below.
>
> **Q:** Could different parenthesizations impact task performance or complexity?
>
> **A:** Different binary trees can in principle change both inductive bias and compute/memory tradeoffs. We use the Blelloch (balanced) tree because it (i) achieves O(log⁡ n) parallel depth, (ii) admits a simple binary-counter decoder with O(log⁡ n) state and work per step, and (iii) naturally allocates more computation to distant context, matching the recency bias of many language tasks. In contrast, left-to-right parenthesization does not support log-depth parallelization for general updates, and arbitrary per-sequence or random trees need not admit any memory-efficient streaming decoder. Alternative fixed trees (e.g., skewed or task-structured hierarchies) could emphasize particular long-range dependencies, but would need to encode non-trivial structure in the data to outperform the balanced choice. We therefore adopt the balanced scan as a conservative, hardware-friendly default and leave task-specific or learned trees to future work.
>
> **Q:** What causes the sensitivity to chunk size?
>
> **A:** Chunk size c controls the tradeoff between local fidelity and how aggressively we compress distant context. Small c yields many fine-grained chunks and a deeper tree; large c yields fewer, coarser chunks and a shallower tree. In the extreme c=n, the tree collapses and Transformer-PSM reduces to a standard Transformer, which is exactly the convergence in perplexity toward the vanilla Transformer curve observed in Figure 5. The optimal c thus depends on how compressible long-range context is for a given task.
>
> **Q:** For large chunk sizes (e.g., c=256), does the memory scaling still hold, or does the chunk-local KV cache in Inf dominate?
>
> **A:** We analyze the dependence on c in Appendices C–D. At inference, per-layer activation memory is O(c(log⁡(n/c)+1)): PSM stores at most log⁡(n/c) prefix chunks (one per tree level) plus a single active chunk in the Inf module. The chunk-local KV cache therefore contributes only an additive O(c) term, and the dominant term remains the log⁡(n/c) prefix states. The O(c log⁡(n/c)) scaling continues to hold even for larger c; when c=n, log⁡(n/c)=0 and the model reduces to a standard Transformer with O(n) memory, consistent with Figure 5.
>
> **Weakness 1:** Are PSM’s advantages (e.g., length generalization on S5) inherent to the prefix-scan framework, or specific to softmax attention?
>
> **A:** On S5 state-tracking tasks, a standard GPT-2 Transformer does not show the same length generalization, suggesting that the key benefit comes from the PSM’s log-depth tree and hierarchical aggregation rather than softmax attention per se. Any learnable function that maps two chunks of tokens to one chunk is a valid PSM aggregator, including simple MLPs, SSM layers, or alternative attention mechanisms. Systematically exploring how such choices affect length generalization is an interesting direction for future work.
>
> **Weakness 2:** The paper does not quantify the tradeoff between chunk size, compute (e.g., $O(c^2)$ for chunk-local attention), and memory (e.g., O(c log⁡(n/c))).
>
> **A:** Appendices C–D derive these tradeoffs for sequence length n, chunk size c, and $L_{agg}$​ and $L_{inf}$​ aggregation and inference layers respectively. Training compute scales linearly in c and n as
> O(cn$L_{agg}$ + cn$L_{inf}$)
>
> (Eq. C1), and per-layer inference memory as
>
> O(c$L_{inf}$ + c log⁡(n/c) $L_{agg}$)
>
> (Eq. C2).  We will move a concise version of this analysis (with equations and a small table) into the main text to make the tradeoffs more visible.
>
> **Weakness 3:** Does the simple embedding Enc limit performance?
>
> **A:** In our design of Transformer-PSM we opted to make the simplest architectural choices, Enc is a standard embedding layer, Agg and Inf are just Transformer layers.  It’s a very interesting question to vary the choice of Enc module.  It is entirely possible that any of a variety of more sophisticated encodings could achieve a spread of performance characteristics superior to a simple but canonical nn.embedding layer.
>
> We would like to thank the reviewer for suggestions on further baselines and tackling additional tasks.  We are actively running these evaluations and hope to share results before the end of the rebuttal period!  We would greatly appreciate it if such updates could be taken into account.
>
> We’d also like to thank the reviewer for their detailed review and questions about our work.  We hope our response has addressed their questions and clarified key aspects of our work.

---

### Official Review · Reviewer_8wg9 · 2025-11-04

**Soundness:** 3
**Presentation:** 3
**Contribution:** 4
**Rating:** 4
**Confidence:** 3

**Summary:**

This paper introduces an algorithmic framework, namely "*Prefix-Scannable Models*" (PSMs), to characterize and generalize sequence models that support both parallelizable training and efficient sequential inference. The authors term this property "Sequential-Parallel Duality" (SPD) and formalize it with the notation SPD(T(n), m(n)), capturing the training compute $T(n)$ and inference memory $m(n)$.
The authors first unify a large class of recent efficient models (e.g., Mamba, GLA, DeltaNet, etc.) under this definition. They show that these models are all instances of an associative prefix scan based on a shared "affine state update". Because their aggregation operator is associative, they can achieve SPD(n, 1) compared to the SPD($n^2, n$) of Transformers. Afterwards, to generalize the framework to non-associative aggregation operators (such as softmax attention), the classic Blelloch parallel prefix scan is used for training, which fixes a specific binary-tree parenthesization. The crucial insight is proposing an online "binary counter" algorithm for inference (Alg. 2) that exactly reproduces this same parenthesization, guaranteeing training-inference consistency. This generalization creates a new class of models, including the proposed *Transformer-PSM*, that achieve SPD(n, log n) with $O(1)$ amortized inference time. Experiments show that Transformer-PSM achieves very good length generalization on state-tracking tasks, and at the same time matching the performance of a full Transformer on associative recall and language modelling.

**Strengths:**

1. **Methodological and technical contributions.** I think that the clean, algorithmic lens it provides for understanding and unifying the recent "zoo" of efficient sequence models is a significant conceptual contribution to the field. Formalizing the SPD(T, m) property, unifying Linear RNNs, SSMs and Transformers, and the transition from associative to non-associative operators is a non-trivial and useful generalization. Technically, using the online binary counter (Alg. 2) to exactly match the Blelloch scan's fixed parenthesization is another instance of applying a classic algorithm to solve a modern deep learning problem.

2. **Empirical evaluations**. The proposed Transformer-PSM model shows very good length generalization result on the $S_5$ state-tracking task (Figure 3), up to sequences >6x longer than the longest training sequence. Furthermore, the MQAR result in Figure 4 show that the Tranformer-PSM can match the recall power of a full Transformer, while the WikiText-103 experiment (Figure 5) shows the trade-off between chunk size and perplexity on language modelling, validating the model's ability to achieve RNN-like efficiency and Transformer-like expressivity on these tasks.

3. **Clarity and Structure**: The paper is well-written, and the use of formal definitions (Def 2.5 for SPD, Def 3.1 for PSM) and algorithms (Alg 1, 2, 3, 4) make the core ideas succinct and easy to follow.

**Weaknesses:**

1. **Limited scale**: The experiments are on small-to-medium datasets (WikiText-103) with relatively small models. While the synthetic results are interesting, it remains to be verified how the architectural benefits and trade-offs (especially $O(\log n)$ memory) of Transformer-PSM scale to state-of-the-art compute with billions of parameters.

2. **Missing baselines and other related work**: While the theoretical unification is broad, the baselines used in the empirical comparisons are narrow, focusing almost exclusively on GPT-2 and Mamba. I think the paper would greatly benefit if the authors would also compare to several highly relevant lines of concurrent work such as:
   - *Transformer-SSM Hybrids*: Models like Jamba, Zamba, etc., which interleave Transformer and SSM layers.
   - *Log-Linear Attention*: The paper cites "Log-linear attention" (Guo et al., 2025) as concurrent work that "fits this view" but provides no empirical comparison, even though it appears to be a direct competitor in the SPD(n, log n) class.
   - *Parallel Non-linear RNNs*: Other approaches to parallelizing non-linear recurrences, such as those based on fixed-point iteration (e.g., DEER and ELK), offer a completely different algorithmic approach to the same problem (parallelizing non-linearity) and should be discussed or benchmarked.
   - *Linear RNNs with enhanced expressivity*: Methods such as DeltaProduct, that have been shown to work great on state-tracking tasks too.

**Questions:**

1. How does Transformer-PSM compare to recent hybrids which alternate Transformer and SSM layers? Do such models fall in the SPD framework too? What are the hypothesized trade-offs in terms of expressivity and parallelization between these two different ways of combining Transformers and efficient recurrences?

2. You successfully parallelize a non-linear operator by restricting its dependencies to a fixed binary-tree structure. Concurrent work like DEER and ELK provides an algorithmic alternative for parallelizing general non-linear RNNs via fixed-point iteration. Could you compare these two fundamental approaches?

3. I would be interested to see more results on other state-tracking tasks such as the ones used in Grazzi et al. 2025. Can the authors provide some results on those tasks, e.g. parity, modular arithmetic w/ and w/o brackets? Additionally, can you also evaluate Mamba2 and DeltaProduct with negative eigenvalues in the state-transition matrix in your experiments?

**-- References --**

Siems et al. Deltaproduct: Improving state-tracking in linear rnns via householder products. In NeurIPS 2025.

Grazzi et al. Unlocking state-tracking in linear rnns through negative eigenvalues. In ICLR 2025.

Gus et al. Log-linear attention. Preprint arXiv:2506.04761, 2025.

AI21 Labs. Jamba: A Hybrid Transformer-Mamba Language Model. arXiv 2024

Glorioso et al. Zamba: A Compact 7B SSM Hybrid Model. arXiv, 2024

Lim et al. Parallelizing non-linear sequential models over the sequence length. In ICLR 2024

Gonzalez et al. Towards Scalable and Stable Parallelization of Nonlinear RNNs. In NeurIPS 2024

---

> ### Author Response · Authors · 2025-11-24
>
> We appreciate the reviewer's thorough and insightful review, along with their positive feedback regarding our work’s significance, novelty and theoretical contribution. In response to their review, we have endeavored to address their concerns as concretely as possible.
>
> **Q:** How does Transformer-PSM compare to recent hybrids which alternate Transformer and SSM layers?
>
> **A:** Our SDP theory is primarily defined for single layers. Nevertheless, it would be easy to define a broader notion that also includes depth and hybrid choices of layers.  The computational complexity of training and the memory bound of inference are still mathematically well defined. By denoting the sequence length N as the dominant scaling factor, a hybrid layer would still incur O($N^2$ * number of Transformer layers + N * number of SSM layers) computational complexity.  The memory would be O(N * number of Transformer layers + c * number of SSM layers) where the hidden state of the SSM is a constant c.  For N large, the PSM would have substantially cheaper computational complexity during training O(N) akin to if the hybrid architecture were comprised entirely of SSM layers.  During inference, the memory bound of the PSM would be O(log N), again akin to if the hybrid architecture were comprised entirely of SSM layers (up to log factors).  As the number of Transformer layers increase in the hybrid architecture, computational complexity of training increases, memory bound at inference increases but expressivity in tasks involving precise recall would also likely improve.
>
> **Q:** Concurrent work like DEER and ELK provides an algorithmic alternative for parallelizing general non-linear RNNs via fixed-point iteration. Could you compare these two fundamental approaches?
>
> **A:** DEER and ELK “Towards Scalable and Stable Parallelization of Nonlinear RNNs” Gonzalez et. al represents a very different approach: they formulate computing the feedforward of an RNN as solving a fixed point operation find $s_0,s_1,...,s_T$ satisfying
> $s_{t+1} = f(s_t)$.
> The number of steps to convergence in theory is stated in their proposition 1, in section 3.2 of their paper, this requires in the worst case O(T) steps which is the same as just feeding the RNN forward.  But the advantage of their formulation is that they may be able to approximately solve for $s_0, …, s_T$ that approximately satisfy $s_{t+1} = f(s_t)$ which is the primary subject of their paper.  In general, there is no way to feed forward a general recurrence forward from its starting input and hidden state in parallel O(1) time.  But it may be possible to do so within a tolerable error, whilst avoiding numerical instabilities which the authors detail in their section 6 limitations.
> Finally, we will add the corresponding discussion and cite these references. Thank you for drawing our attention to them.
>
> **Weakness:** Limited scale...
>
> We agree that validating PSMs at billion-parameter scale and ultimately in industrial workloads is an important next step, but this is unfortunately beyond our current compute budget. Our experiments therefore focus on “academic-scale” models where we can systematically control for architecture and data; importantly, the proposed kernels are architecture- and hardware-agnostic, so they can be dropped into larger training runs without changing the algorithmic guarantees (SPD, O(n) work, and O(log⁡n) depth).
>
> We would like to thank the reviewer for suggestions on further baselines and tackling additional tasks.  We are actively running these evaluations and hope to share results before the end of the rebuttal period!  We would greatly appreciate it if such updates could be taken into account.
>
> We’d also like to thank the reviewer for their detailed review and questions about our work.  We hope our response has addressed their questions and clarified key aspects of our work.

---

### Meta-Review · Area_Chair_PzFD · 2026-01-11

**Summary:**

This paper proposes an algorithmic framework to characterize and generalize neural sequence models that can be trained in highly parallel fashion while allowing fast sequential decoding with small memory. The reviewers broadly agree the paper is sound and clearly presented, with a notable conceptual contribution: it provides a principled, algorithmic framework that unifies existing efficient architectures and introduces a credible path to extend “scan-based” efficiency to attention-like mechanisms while keeping decoding efficient and consistent with training. After carefully reading the paper, review and author responses, the AC agrees with the majority of the reviewers on accepting the paper.

**Reviewer Concerns:**

see Summary

**Reviewer Scores:**

see Summary

---

### Decision · Program_Chairs · 2026-01-26

Accept (Poster)